# A feature of maternal sleep apnea during gestation causes autism-relevant neuronal and behavioral phenotypes in offspring

**Amanda M. Vanderplow[1], Bailey A. Kermath[1], Cassandra R. Bernhardt[1], Kimberly T. Gums[1], Erin N. Seablom[1], Abigail B. Radcliff[1], Andrea C. Ewald[1], Mathew V. Jones[2], Tracy L. Baker[1], Jyoti J. Watters[1], Michael E. Cahill[1]***

**1** Department of Comparative Biosciences, University of Wisconsin-Madison, Madison, Wisconsin, United States of America, **2** Department of Neuroscience, University of Wisconsin-Madison, Madison, Wisconsin, United States of America

* michael.cahill@wisc.edu

**Data Availability Statement:** All relevant data are within the paper and its Supporting Information files.

## Abstract

Mounting epidemiologic and scientific evidence indicates that many psychiatric disorders originate from a complex interplay between genetics and early life experiences, particularly in the womb. Despite decades of research, our understanding of the precise prenatal and perinatal experiences that increase susceptibility to neurodevelopmental disorders remains incomplete. Sleep apnea (SA) is increasingly common during pregnancy and is characterized by recurrent partial or complete cessations in breathing during sleep. SA causes pathological drops in blood oxygen levels (intermittent hypoxia, IH), often hundreds of times each night. Although SA is known to cause adverse pregnancy and neonatal outcomes, the long-term consequences of maternal SA during pregnancy on brain-based behavioral outcomes and associated neuronal functioning in the offspring remain unknown. We developed a rat model of maternal SA during pregnancy by exposing dams to IH, a hallmark feature of SA, during gestational days 10 to 21 and investigated the consequences on the offspring's fore-brain synaptic structure, synaptic function, and behavioral phenotypes across multiples stages of development. Our findings represent a rare example of prenatal factors causing sexually dimorphic behavioral phenotypes associated with excessive (rather than reduced) synapse numbers and implicate hyperactivity of the mammalian target of rapamycin (mTOR) pathway in contributing to the behavioral aberrations. These findings have implications for neuropsychiatric disorders typified by superfluous synapse maintenance that are believed to result, at least in part, from largely unknown insults to the maternal environment.

## Introduction

Many psychiatric disorders originate from a complex interplay between genetics and early life experiences, particularly in the womb [1]. Mounting epidemiologic and scientific evidence indicates that maternal immune activation during pregnancy is a key trigger for neurodevelopmental deficits in offspring [2–8]. One common inflammatory stimulus that may be

**Funding:** This work was supported by the following sources: National Institute of Health (NIH R01 HL 142752) (to TLB and JJW) https://www.nih.gov/ The funders had no role in study design, data collection and analysis, decision to publish, or preparation of the manuscript NARSAD Young Investigator Award (28303) from Brain & Behavior Research Foundation (to MEC) https://www.bbrfoundation.org/ The funders had no role in study design, data collection and analysis, decision to publish, or preparation of the manuscript University of Wisconsin at Madison UW2020 award (to TLB and JJW) https://research.wisc.edu/funding/uw2020/ The funders had no role in study design, data collection and analysis, decision to publish, or preparation of the manuscript.

**Competing interests:** The authors have declared that no competing interests exist.

**Abbreviations:** cDNA, complementary DNA; CV, chevron; DiI, 1,1′-dioctadecyl-3,3,3′,3′-tetramethylindocarbocyanine perchlorate; DLPFC, dorsolateral prefrontal cortex; DN, down; E19, embryonic day 19; FS, frequency step harmonic; GIH, gestational intermittent hypoxia; GNX, gestational normoxia; HIF-1α, hypoxia-inducible factor 1α; HM, harmonic; IH, intermittent hypoxia; IL, interleukin; mPFC, medial prefrontal cortex; MS, mixed syllable; mTOR, mammalian target of rapamycin; MX, mixed; OT, other; P4, postnatal day 4; P10, postnatal day 10; PFA, paraformaldehyde; SA, sleep apnea; sEPSC, spontaneous excitatory postsynaptic current; ST, short; TNFα, tumor necrosis factor alpha; USV, ultrasonic vocalization.

experienced during pregnancy is maternal sleep apnea (SA) [9]. SA is characterized by recurrent partial or complete cessation of breathing during sleep, often hundreds of times each night. The resulting swings in blood oxygen levels (intermittent hypoxia, IH) induce profound inflammation, which causes most of the morbidities associated with SA [10–13]. The incidence of SA during pregnancy is on the rise worldwide in line with the obesity epidemic and is estimated to occur in approximately 15% of normal and >60% of high-risk pregnancies by the third trimester [14–17]. It is largely appreciated that maternal SA during pregnancy is detrimental to the health of the pregnancy and the newborn [10,18–28]. Absent, however, is an understanding of whether maternal SA has long-term detrimental consequences on her offspring or if maternal SA impacts offspring neural development.

Intriguing correlative data are consistent with the idea that maternal SA during pregnancy may lead to long-lasting alterations in offspring neural function in humans. Indeed, the same pregnancy complications known to directly result from maternal SA are all risk factors for the later development of psychiatric disorders in the offspring. For example, SA during pregnancy is most prevalent in women that are obese, have gestational hypertension, or are of advanced age. In addition, maternal SA during pregnancy increases risk for gestational diabetes, pre-eclampsia, preterm birth, fetal growth restriction, and neonatal intensive care admission [19,21,26,29–33]. Accumulating evidence indicates that these risk factors for, and consequences of, maternal SA are associated with the development of offspring neural disorders later in life such as autism spectrum disorder, attention-deficit/hyperactivity disorder, intellectual disability, and schizophrenia [34–40].

Based on clinical correlations, a recent medical hypothesis paper theorized that maternal SA could cause the development of psychiatric disorders in the offspring and, in particular, lead to autism spectrum disorders [41]. However, no animal model study has systematically investigated a potential link between maternal SA and aberrant neuronal and behavioral outcomes in her offspring. Indeed, before such a link can be systematically investigated in humans, animal models must necessarily provide the justification. Further, as ethical concerns preclude allowing a pregnancy to proceed in a mother with SA without medical intervention, no large-scale prospective studies have investigated links between maternal gestational SA and offspring neural and behavioral phenotypes. However, a recent retrospective epidemiological study found that male children whose mothers had SA during pregnancy had an increased risk of "developmental vulnerability," with lower cognitive testing outcomes and increased special needs status [42]. Although this study was limited by a small sample size and was not sufficiently powered to detect links to specific neuropsychiatric disorders, to our knowledge, it represents the first indication that maternal SA during pregnancy can have sexually dimorphic effects on neurodevelopmental outcomes in human offspring.

To begin to understand the impact of maternal SA on offspring neural function, we developed a rat model by exposing pregnant rats dams to IH during their sleep cycle in the latter part of pregnancy, the time point where most pregnant women develop SA [33]. IH is a hallmark feature of SA and is frequently used to model SA in animal models as it replicates many of the core consequences of SA in humans [43]. We then investigated the consequences on offspring forebrain synaptic structure, synaptic function, and behavioral phenotypes across multiples stages of postnatal development. For the first time, our results identify maternal IH during gestation as a potent inducer of cortical hyperconnectivity and hyperfunction in the offspring, with far-ranging behavioral aberrations that span communicative, cognitive, and social domains. Importantly, the magnitude and breadth of these alterations are consistently more severe in male than female offspring. Finally, our findings implicate hyperactive mammalian target of rapamycin (mTOR) signaling in the cortex as a major contributor to the impaired cognitive and social behaviors in the male offspring. Collectively, these data indicate

that maternal SA, and the resulting chronic exposure to IH, may be an unrecognized trigger for neurodevelopmental disorders in her offspring and further highlight the therapeutic potential of dampening mTOR signaling in conditions typified by excessive synaptic connectivity.

## Results

Pregnant rat dams were exposed to IH from gestational days 10 through 21 (gestational intermittent hypoxia, GIH). IH consisted of cyclic bouts of 2 minutes of hypoxia (10.5% oxygen) followed by 2 minutes of normoxia (21% oxygen) over the course of 8 hours per day during their typical period of heightened sleep. These parameters were designed to mimic the desaturation and reoxygenation kinetics in humans with SA, and pregnant dams were exposed during this period of gestation as the risk for the development of gestational SA in humans more than doubles after the first trimester [33]. In order to prevent any direct exposure of the offspring to hypoxia, on gestational day 21, exposure to the cyclic oxygen fluctuations ceased. As a control, pregnant dams were subjected to the exact same conditions as in GIH, including placement in the environmental chamber and exposure to flowing gases, but oxygen oscillated at normal levels of 21% and 21% (gestational normoxia, GNX). GIH did not affect gestation time, litter size, or litter survival, indicating that GIH did not have any gross effects of the health of the pregnancy. Further, we found normal weight growth of male and female GIH offspring pups while in the care of their mothers during preweaning ages (through 3 weeks of age) as male GIH offspring pups were within 4% of GNX males at all time points [2-way repeated measures ANOVA $F_{(2,40)} = 0.98$, $p = 0.38$; $n = 10$ GNX, 12 GIH] and female GIH offspring pups within 2% of GNX females at all time points [2-way repeated measures ANOVA $F_{(2,58)} = 0.47$, $p = 0.63$; $n = 14$ GNX, 17 GIH]. With the exception of oxygen desaturation validation experiments in pregnant rat dams, offspring from GIH and GNX mothers were used for all experiments.

First, we validated that oxygen desaturation occurs in a subset of pregnant rat dams subjected to the cyclic oxygen fluctuations of GIH detailed above. Using pulse oximetry during the hypoxic episodes, in 5 pregnant dams, oxyhemoglobin saturation reached a nadir of 78% at the peak of hypoxia (**S1A and S1B Fig**). These findings confirm that the GIH procedure causes oxygen desaturation in pregnant dams. Pups from these dams were not included in subsequent experiments to avoid potential confounding effects of maternal stress associated with the placement of the pulse oximeter collar.

Much evidence indicates that via the placenta, the fetus is largely protected from the direct effects of maternal hypoxia [44,45]. In this way, intermittent maternal hypoxia during pregnancy is not expected to cause a concomitant hypoxic state in the fetus. As our GIH model was developed to specifically evaluate the effects of maternal hypoxia during gestation on offspring neural outcomes, we wanted to confirm the lack of a hypoxic state in the fetus in our model. To this end, we investigated the gene expression of the hypoxia-inducible factor 1α (HIF-1α), which is robustly up-regulated in response to cellular hypoxia as a means to increase oxygen transport and anaerobic ATP production. As such, changes in organ levels of *HIF-1α* are a commonly used, validated, and sensitive indicator of hypoxia, including those in the brain [46,47]. Despite decreases in arterial oxygen saturation in the pregnant dam to approximately 78% with each hypoxic episode, no changes in *HIF-1α* in either the placenta or in fetal brains from GIH pregnancies were detected relative to GNX (**S1C and S1D Fig**). Further, injection of pregnant dams with an oxygen sensing probe that forms protein thiol adducts in hypoxic cells [48] failed to detect changes in GIH placentas or fetal brains relative to GNX (**S1E–S1H Fig**). These data are consistent with our GIH procedure producing an expected normoxic state in the fetus.

To determine if maternal GIH has functional consequences on offspring behavior, a battery of behavioral assessments that span cognitive, social, and affective domains were performed in GIH and GNX offspring. Rats from a minimum of 3 GNX and GIH litters (1 litter per mother) were used per behavioral assessment, and the *n* for statistical analyses was derived from the total number of offspring across litters per condition. Behavioral assessments were performed throughout numerous developmental periods, including juvenile stages, adolescence, and into adulthood.

## Juvenile GIH offspring exhibit alterations in ultrasonic vocalizations

Most cognitive, social, and affective behavioral assessments cannot be reliably tested in rats during preweaning ages. However, rat vocal communication, referred to as ultrasonic vocalization (USV), emerges during early juvenile development. These USVs can be reliably elicited and measured in rat pups in response to brief separation from their mothers and are thought to represent a distress call to obtain maternal attention [49–51]. After home cage maternal separation, postnatal day 4 (P4) pups were transferred to sound-attenuated isolation chambers, and USVs emitted over a 90-second period in male and female offspring of GIH and GNX mothers were analyzed (**Fig 1A**). For total USVs emitted at P4, we found a main effect of maternal GNX/GIH status [F (1,40) = 18.90, *p* < 0.0001], and while GNX offspring reliably emitted USVs during maternal separation, the total number of USVs was significantly increased in male and female GIH offspring (**Fig 1B**). The vocalizations elicited by rat pups can be divided into numerous distinct categories, which at P4 include harmonic (HM), mixed syllable (MS), frequency step harmonic (FS), short (ST), chevron (CV), up, and down (DN) (**Fig 1C**). When assessing the elicitation of these different USV categories in P4 male pups, we found a main effect for maternal GIH/GNX status [F (1,140) = 4.727, *p* = 0.0314]; however, no differences among individual call types were identified between GNX and GIH offspring (**Fig 1D**). In P4 female pups, we also identified a main effect for maternal GIH/GNX status [F(1,140) = 5.523, *p* = 0.0202], and post hoc analyses identified a significant increase in the frequency of HM and FS call types in GIH versus GNX female offspring (**Fig 1E**). No changes in other call parameters, including mean call duration and mean peak frequency, were detected between groups in P4 males or females (**S2A–S2D Fig**).

To determine if the increased elicitation of USVs persists into subsequent juvenile stages, we also measured maternal separation–induced USVs in male and female offspring of GIH and GNX mothers at postnatal day 10 (P10) (**Fig 1F**). Similar to P4, for total USVs at P10, we detected a main effect for maternal GNX/GIH status [F(1,41) = 27.01, *p* < 0.0001] with a significant increase in the total number of USVs in male and female GIH offspring as compared to sex-matched GNX offspring (**Fig 1G**). At P10, we assessed HM, ST, and mixed (MX) call categories, as well as calls that do not fit into these classifications (referred to as "other"—OT) (**Fig 1H**). Call category analysis revealed a main effect for GNX/GIH maternal status in both male and female offspring [male: F(1,80) = 12.98, *p* = 0.0005; female: F(1,84) = 8.714, *p* = 0.0041]. Further, in male P10 GIH offspring, we detected a significant increase in the emittance of HM and MX calls relative to GNX offspring (**Fig 1I**), while in female P10 GIH offspring, a significant difference was detected only for MX calls (**Fig 1J**). No differences in other call parameters were detected between groups in either male or female P10 rats (**S2E–S2H Fig**). Collectively, these findings indicate that juvenile GIH offspring, regardless of sex, emit a greater number of USVs in response to maternal separation than GNX offspring.

## GIH impairs cognitive function in adolescent male, but not female, GIH offspring

Next, we assessed cognitive behavior in adolescent offspring of GIH and GNX mothers (**Fig 2A**). Spontaneous alternation behavior in the Y-maze, a well-validated index of spatial

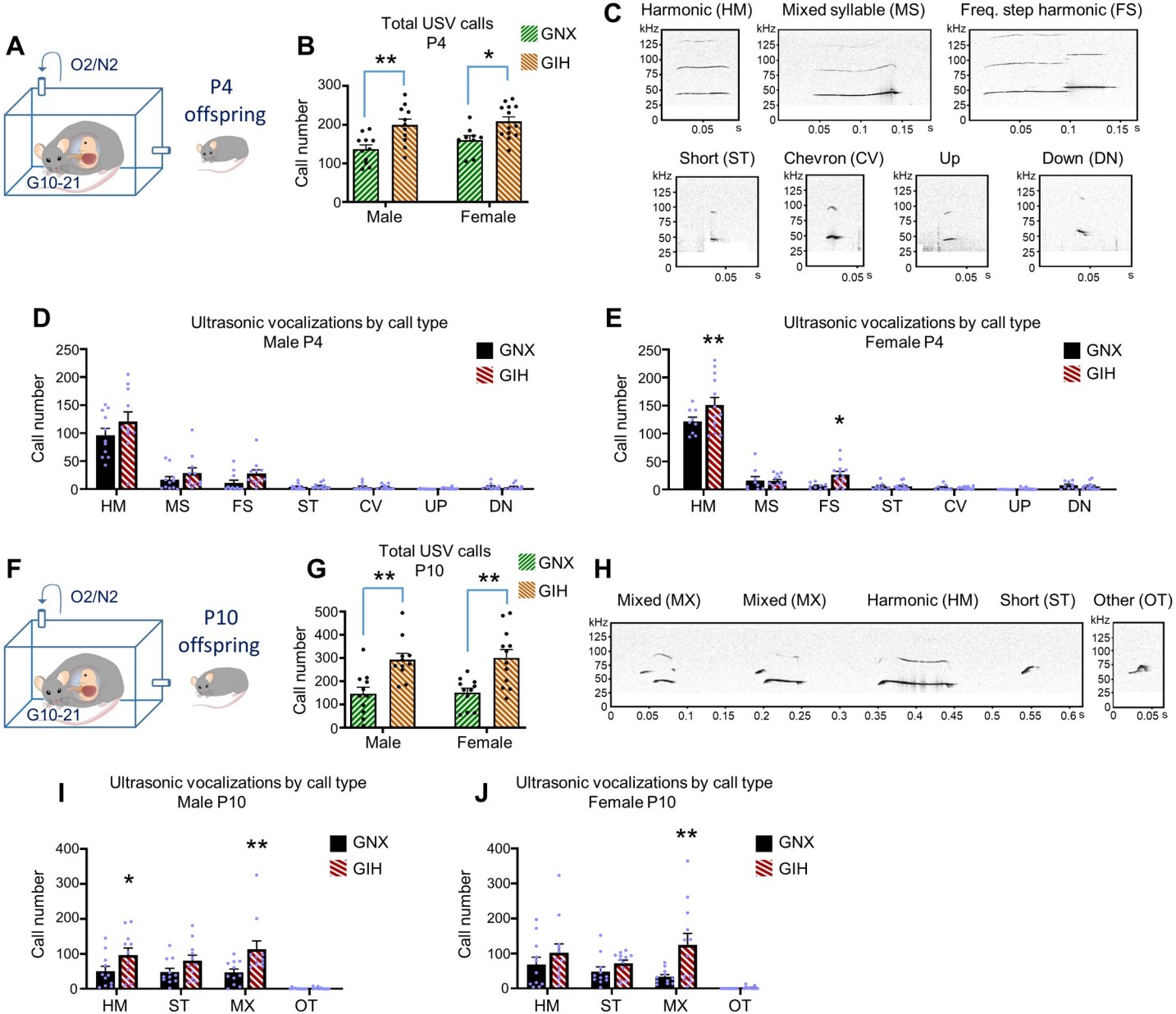

**Fig 1. USVs in juvenile offspring of GIH mothers are increased.** (A) Schematic depicting GNX and GIH offspring age for data pertaining to **Fig 1B–1E**. (B) Total USV calls over a 90-second period in P4 male and female GNX and GIH rat offspring. A significant increase in call number was detected in male GIH [Bonferroni post hoc (df,40), $p$ = 0.0023] and female GIH offspring [Bonferroni post hoc (df,40) $p$ = 0.0232] versus sex-matched GNX counterparts. $n$ = 11 male GNX, 11 male GIH, 9 female GNX, 13 female GIH rats. (C) Sample traces of USV subtypes elicited in P4 rats in response to maternal separation. (D) Mean USV subtype calls over a 90-second period in P4 male GNX and GIH offspring rats. No differences in any subtype calls were detected between groups. [False discovery correction post hoc (df, 140): HM, q = 0.082; MS, q = 0.492; FS, q = 0.301; ST, CV, UP, and DN, q = 1.0] $n$ = 11 GNX and 11 GIH rats. (E) Mean USV subtype calls over a 90-second period in P4 female GNX and GIH offspring rats. GIH offspring exhibited a significant increase in number of HM [false discovery correction post hoc (df, 140) q = 0.0013] and FS [false discovery correction post hoc (df,140) q = 0.0231] calls relative to GNX offspring [MS, ST, CV, UP, and DN (df,140) q = 0.7302]. $n$ = 9 GNX and 13 GIH rats. (F) Schematic depicting offspring age for data pertaining to **Fig 1G–1J**. (G) Total USV calls in P10 male and female GNX and GIH offspring rats over a 90-second period in response to maternal separation. A significant increase in call number was detected in male [Bonferroni post hoc (df,41) $p$ = 0.0017] and female [Bonferroni post hoc (df,41) $p$ = 0.0011] GIH offspring versus sex-matched GNX counterparts. $n$ = 11 male GNX, 11 male GIH, 11 GNX female, 12 GIH female rats. (H) Sample traces of USV subtypes in P10 rats in response to maternal separation. (I) Mean USV subtype calls over a 90-second period in P10 male GNX and GIH offspring rats. GIH offspring exhibited a significant increase in the number of HM [false discovery correction post hoc (df,80) q = 0.0392] and MX [false discovery correction post hoc (df,80) q = 0.0051] calls relative to GNX offspring [ST (df,80) q = 0.1167; OT (df,80) q = 0.7623]. $n$ = 11 GNX and 11 GIH rats. (J) Mean USV subtype calls over a 90-second period in P10 female GNX and GIH offspring rats. GIH offspring exhibited a significant increase in the number of MX calls [false discovery correction post hoc (df,84) q = 0.0016] relative to GNX offspring [HM (df,84) q = 0.3065; ST (df,84) q = 0.3953; OT (df,84) q = 0.7299]. $n$ = 11 GNX and 12 GIH rats. All bar graphs are the mean + SEM. The data underlying this figure can be found in S1 Raw Data. CV, chevron; DN, down; FS, frequency step harmonic; GIH, gestational intermittent hypoxia; GNX, gestational normoxia; HM, harmonic; MS, mixed syllable; OT, other; P4, postnatal day 4; P10, postnatal day 10; ST, short; USV, ultrasonic vocalization.

working memory, was assessed (**Fig 2B**). This test relies on the natural tendency of rats to explore novelty such that rats will typically explore a given arm of the maze just once per every third arm entry [52–54]. Further, this test avoids exposing animals to inherently stressful stimuli (e.g., water) as done in other working memory tasks. Y-maze arm entries were monitored over a 10-minute period in male and female offspring, and the percentage of successive overlapping arm entry triplets was assessed. This task requires working memory as rats must continually maintain and update a mental choice log of arm entry order [55]. As reentry into an arm that a rat just exited is rare, chance performance in this Y-maze task is 50% alternation. When assessing GIH versus GNX offspring performance by sex, we found a significant main effect of GNX/GIH status on spontaneous alternation performance [F(1,69) = 7.139, $p$ = 0.0094]. Post hoc analyses indicated that male, but not female, GIH offspring rats exhibited an alternation deficit relative to their sex-matched GNX offspring counterparts, with further analysis indicating the male GIH offspring failed to perform above chance levels (**Fig 2C**).

While working memory in rodents represents a transient and flexible holding in mind of information for brief periods of time enabling this information to undergo continual updating [56,57], various forms of recognition memory in rodents represent more stable, persistent maintenance of information (e.g., hours to days) [58]. To determine if longer-term information maintenance is also impaired in adolescent GIH offspring, we assessed rats in a novel object recognition task in which animals must differentiate familiar objects from novel objects across a 24-hour span [59,60] (**Fig 2D**). The recognition of object novelty manifests by an increased proportion of time investigating novel versus familiar objects during trial 2. Importantly, we found no effects of GIH status on time spent investigating 2 copies of the same object during trial 1 (**Fig 2E**), indicating that the motivation to explore novelty is intact in these rats. For trial 2, a recognition index was calculated as the percentage of total time spent exploring objects that was devoted to the novel object during the second trial; chance level = 50%. Analysis of recognition index revealed main effects for sex [F(1,37) = 17.92, $p$ = 0.0001] and GIH/GNX status [F(1,37) = 17.53, $p$ = 0.0002]. Further, post hoc analyses indicated that while male GIH offspring showed a significantly reduced recognition index relative to male GNX offspring, this difference between groups was not detected in female rats (**Fig 2F**).

## GIH impairs social behavior in adolescent male, but not female, GIH offspring

To determine if the behavioral deficits of adolescent GIH offspring rats extended to the social domain, we assessed animals in a 3-chamber social approach and preference for social novelty task [61,62]. The social apparatus consists of 2 end chambers of equal size separated by a transparent divider from a smaller middle chamber. The divider contains openings that allow the animals to freely navigate between chambers. During the habituation phase, rats were familiarized with the layout of the social apparatus. During social approach testing, 2 identical mesh cylinders were placed in each end chamber, with one cylinder containing an unfamiliar sex and age-matched stimulus rat, while the other cylinder remained empty (**Fig 2G**). The amount of time rats spent directly investigating the cylinders in each end chamber was assessed. When comparing the amount of time male GIH and GNX offspring spent investigating the empty cylinder versus the stimulus cylinder, we found main effects for maternal GNX/GIH status [F(1,18) = 7.156 $p$ = 0.0154] and end chamber cylinder [F(1,18) = 137.7, $p$ < 0.0001] and also found a significant interaction between these parameters [F(1,18) = 19.34, $p$ = 0.003]. Further, post hoc testing revealed that while both GNX and GIH male offspring showed a preference for the social cylinder relative to the empty cylinder ($p \leq$ 0.0001 for both groups), the GNX

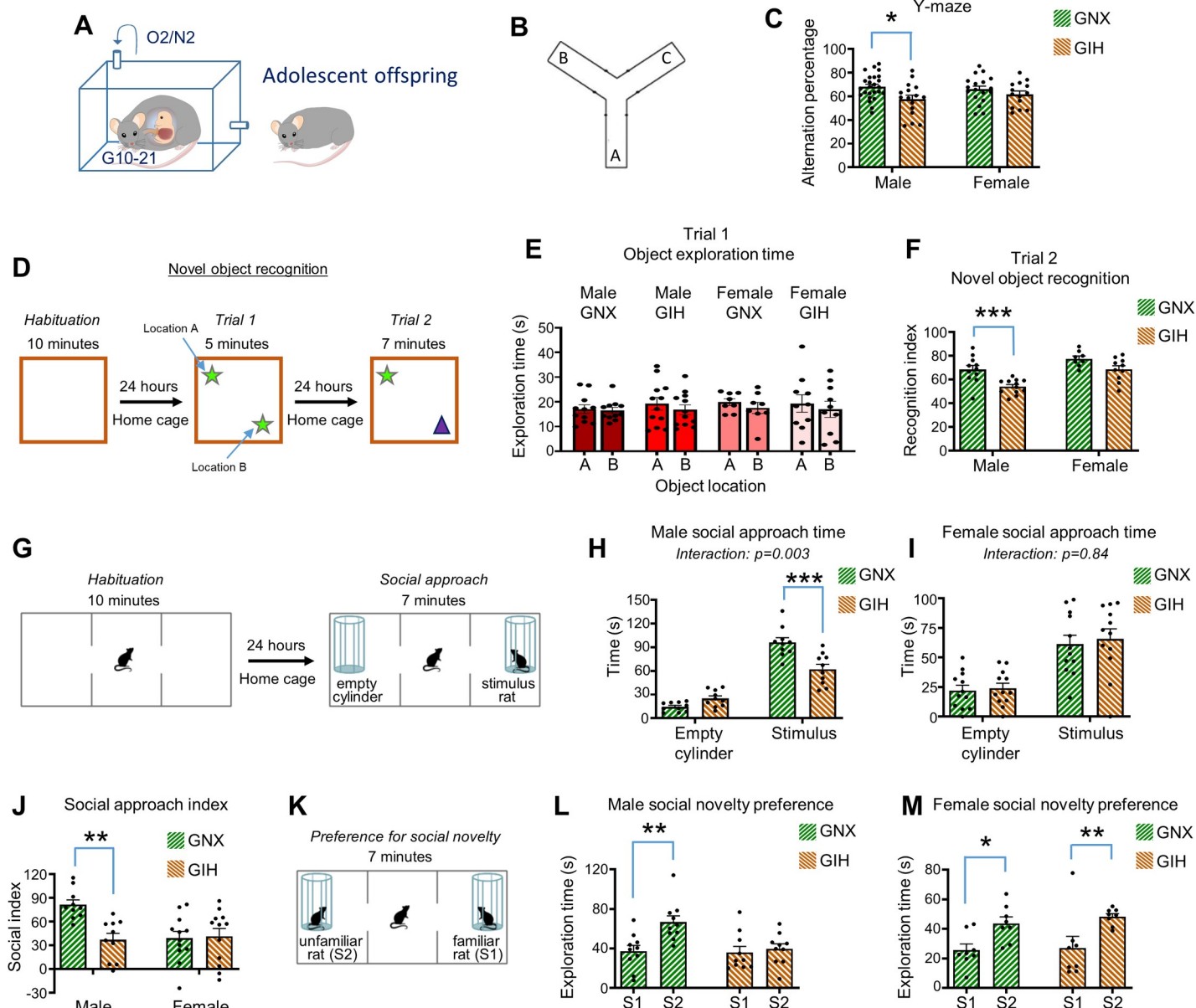

**Fig 2. Male adolescent GIH offspring display cognitive and social phenotypes.** (A) Schematic depicting GNX and GIH offspring age pertaining to all data in **Fig 2**. (B) Y-maze schematic. (C) Mean Y-maze spontaneous alternation percentage. Male [Bonferroni post hoc (df,69) $p$ = 0.0115], but not female (Bonferroni post hoc (df,69) $p$ = 0.6126), GIH offspring show an alternation deficit relative to GNX offspring. $n$ = 25 male GNX, 18 female GNX, 16 male GIH, 14 female GIH rats. (D) Schematic of the novel object recognition task. (E) Object exploration time during trial 1 of the object recognition test revealed no differences in time investigating either object location in GNX and GIH male or female offspring for all group comparisons [Bonferroni post hoc (df,37), male GNX $p$ = 1.0; male GIH $p$ = 0.6596; female GNX $p$ = 1.0; female GIH $p$ = 0.9064]. $n$ = 11 male GNX, 8 female GNX, 12 male GIH, 10 female GIH rats. (F) Novel object recognition index during trial 2 revealed a deficit in male GIH offspring [Bonferroni post hoc (df,37) $p$ = 0.0008] with no significant impairment in female GIH offspring [Bonferroni post hoc (df,37) $p$ = 0.0782]. $n$ = 11 male GNX, 8 female GNX, 12 male GIH, 10 female GIH rats. (G) Schematic of social approach task. (H) Time spent investigating each cylinder in the social approach task for male adolescents. GIH offspring spent less time investigating the stimulus rat cylinder than GNX offspring [Bonferroni post hoc (df,36) $p$ < 0.0001]. $n$ = 10 male GNX, 10 male GIH. (I) Time spent investigating each cylinder in the social approach task for female adolescents. No differences in time spent investigating the social cylinder was detected between GNX and GIH offspring [Bonferroni post hoc (df,44) $p$ > 0.999]. $n$ = 12 female GNX, 12 female GIH. (J) Social approach revealed a significant social index deficit in male GIH offspring [Bonferroni post hoc (df,40) $p$ < 0.002] with no corresponding alteration in female GIH offspring (Bonferroni post hoc (df,40) $p$ = 1.0). $n$ = 10 male GNX, 12 female GNX, 10 male GIH, 12 female GIH rats. (K) Schematic of preference for social novelty task. (L) Social novelty preference in male GIH and GNX offspring. Whereas male GNX offspring exhibited a significant increase in the amount of time investigating the novel (S2) versus familiar rat (S1) [Bonferroni post hoc (df,18) $p$ = 0.0076], male GIH rats did not show an increased preference for social novelty [Bonferroni post hoc (df,18) $p$ = 1.0]. $n$ = 10 GNX and 10 GIH rats. (M) Social novelty preference in female GIH and GNX offspring. Both GNX and GIH female offspring show a significant increase in time spent investigating the novel (S2) versus familiar (S1) rat [Bonferroni post hoc (df,14) GNX $p$ = 0.0242; GIH $p$ = 0.0093]. $n$ = 8 GNX and 8 GIH rats. All bar graphs are the mean + SEM. The data underlying this figure can be found in S1 Raw Data. GIH, gestational intermittent hypoxia; GNX, gestational normoxia; S1, stimulus rat 1; S2, stimulus rat 2.

male offspring spent significantly more time investigating the social cylinder compared to GIH male offspring ($p < 0.0001$) (Fig 2H, S3A–S3E Fig). On the other hand, no interaction between maternal GNX/GIH status and end chamber cylinder was found in female offspring [$F_{(1,22)} = 0.04$, $p = 0.8431$], with both groups showing similar amounts of time exploring the social cylinder (Fig 2I). To more directly compare sex differences in maternal GNX/GIH status on social approach, we calculated an approach index defined as the amount of time investigating the cylinder containing a novel rat minus the time investigating the empty cylinder; chance level = 0. We found main effects for both maternal GNX/GIH status [$F_{(1,40)} = 6.116$, $p = 0.0177$] and sex [$F_{(1,40)} = 5.024$, $p = 0.0306$]. While both male and female GNX offspring exhibited an above chance preference for the novel rat, this preference was considerably lower for GNX female rats as compared to GNX male rats. This is consistent with previous studies showing a reduced social end chamber preference in female rats compared to males using similar methods [63–65]. Importantly, we found that whereas male GIH offspring rats showed a significantly reduced preference for the novel rat as compared to male GNX offspring, no corresponding deficit was present in female GIH offspring rats (Fig 2J).

Following social approach testing, an unfamiliar, age, and sex-matched rat was placed in the mesh cylinder that was previously empty during the approach task. Rats were returned to the apparatus and the amount of time investigating each end chamber was assessed as an index of social novelty preference, as rats can choose between interacting with a familiar or an unfamiliar conspecific (Fig 2K). In males, we found a main effect for maternal GNX/GIH status [$F_{(1,18)} = 6.936$, $p = 0.0169$] and for familiar/novel rat [$F_{(1,18)} = 6.957$, $p = 0.0167$]. Further, whereas male GNX offspring showed an increased preference for the unfamiliar rat versus the familiar rat, male GIH offspring did not exhibit this increased preference for social novelty (Fig 2L). On the other hand, among female rats, while a main effect for familiar/novel rat was identified [$F_{(1,14)} = 19.50$, $p = 0.0006$], there was no main effect of GNX/GIH maternal status [$F_{(1,14)} = 0.2901$, $p = 0.5986$]. Further, female GIH offspring showed a normal level of preference for the novel conspecific (Fig 2M). Taken together, these findings indicate that maternal GIH exposure causes deficits in sociability and preference for social novelty in male, but not female, adolescent GIH offspring.

## Sex-dependent cognitive and social impairments in adolescent GIH offspring persist into adulthood

To determine if the cognitive and social deficits identified in adolescent GIH male offspring persisted into adulthood, we assessed behaviorally naive adult male GNX and GIH offspring (Fig 3A). Like our observations in late adolescents, adult male GIH offspring exhibited significant deficits in Y-maze spontaneous alternation percentage (Fig 3B) and novel object recognition memory (Fig 3C and 3D). For social approach, both adult male GNX and GIH offspring showed a greater preference for the stimulus cylinder relative to the empty cylinder (Bonferroni post hoc, GNX $p < 0.0001$; GIH $p = 0.0023$). Direct comparison revealed a reduction in the amount of time GIH adult male offspring spent investigating the stimulus cylinder compared to GNX adult offspring ($p = 0.03$); however, this difference did not quite survive multiple comparison correction (Bonferroni post hoc, $p = 0.06$). Nevertheless, that we detected a significant interaction between maternal GNX/GIH status and end chamber cylinder [$F_{(1,18)} = 4.829$, $p = 0.0413$] suggests that the preference for the social versus empty cylinder is greater in the GNX group (Fig 3E). This point is further supported by the reduced social approach index in the adult male GIH offspring as compared to GNX offspring (Fig 3F). Adult GIH male offspring also showed a reduced preference for social novelty (Fig 3G). Further, we assessed open field behavior in GNX and GIH male adult offspring (Fig 3H), examining basic movement

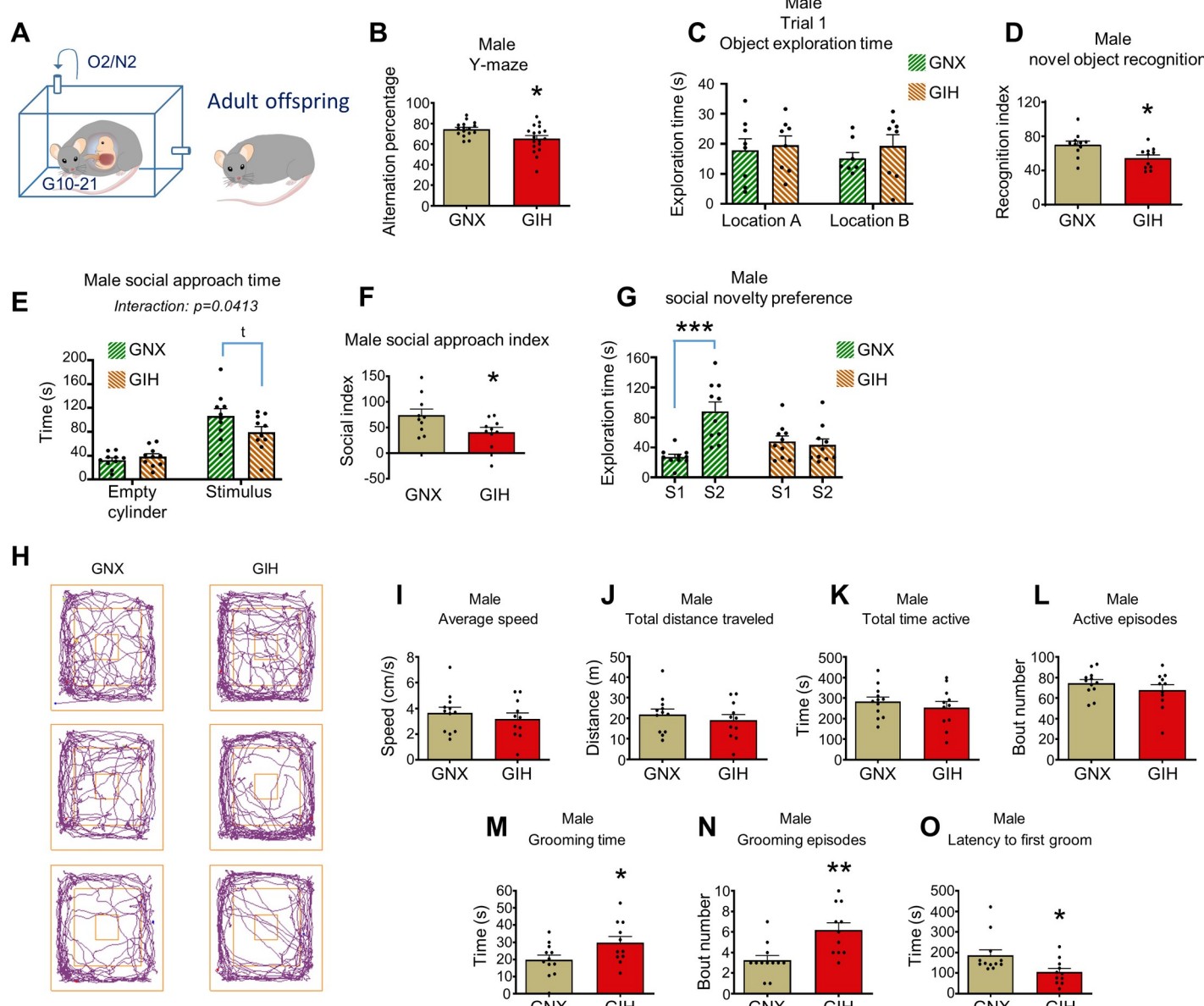

**Fig 3. Behavioral phenotypes persist into adulthood in male GIH offspring. (A)** Schematic depicting GNX and GIH offspring age pertaining to all data in **Fig 3**. **(B)** Mean Y-maze spontaneous alternation percentage. Adult male GIH offspring show an alternation deficit relative to GNX offspring [t (df,32) = 2.541, *p* = 0.0137]. *n* = 17 GNX, 18 GIH rats. **(C)** Object exploration time during trial 1 of the object recognition test reveals no differences in time investigating either object location in GNX and GIH male offspring [Bonferroni post hoc (df,28), location A *p* = 1.0; location B *p* = 0.7534]. *n* = 8 male GNX, 8 male GIH rats. **(D)** Assessment of mean novel object recognition index reveals a deficit in adult male GIH offspring [t (df,20) = 2.749, *p* = 0.0124]. *n* = 11 male GNX, 11 male GIH rats. **(E)** Time spent investigating each cylinder in the social approach task for male adults. Direct comparison reveals that GIH offspring spent less time investigating the stimulus rat cylinder than GNX offspring (*p* = 0.03); however, this difference did not quite survive multiple comparison [Bonferroni post hoc (df,36) *p* = 0.06]. *n* = 10 male GNX, 10 male GIH rats. **(F)** Social approach index assessment revealed a significant deficit in male GIH offspring [t (df,18) = 2.198, *p* = 0.0413]. *n* = 10 male GNX, 10 male GIH rats. **(G)** Social novelty preference analysis indicates that whereas male GNX offspring exhibited a significant increase in the amount of time spent investigating the novel (S2) versus familiar (S1) rat [Bonferroni post hoc (df,18) *p* = 0.0005], male GIH rats did not show an increased preference for social novelty [Bonferroni post hoc (df,18) *p* = 1.0]. *n* = 10 GNX and 10 GIH rats. **(H)** Sample locomotor traces in an open field across 10 minutes for male GNX and GIH offspring. Outer, middle, and inner zones of the open field are shown in orange. **(I–L)** No differences in mean speed in the open field (I), total distance traveled (J), total time active (K), or the number of active episodes (L) were detected between adult male GNX and GIH offspring [(df,21) open field t = 0.7172, *p* = 0.4811; total distance t = 0.7162, *p* = 0.4818; total active t = 0.7837, *p* = 0.4420; active episodes t = 1.071, *p* = 0.2963]. *n* = 12 GNX and 11 GIH rats. **(M)** Adult male GIH offspring showed a significant increase in time spent grooming in the open field compared to GNX offspring [t (df,21) = 2.179, *p* = 0.0408]. *n* = 12 GNX and 11 GIH rats. **(N)** Adult male GIH offspring showed a significant increase in the number of grooming episodes in the open field versus GNX offspring [t (df,21) = 3.473, *p* = 0.0022]. *n* = 12 GNX and 11 GIH rats. **(O)** Adult male GIH offspring showed a shorter latency to the first grooming episode in the open field as compared to GNX offspring [t (df,21) = 2.502, *p* = 0.0207]. *n* = 12 GNX and 11 GIH rats. All bar graphs are the mean + SEM. The data underlying this figure can be found in S1 Raw Data. GIH, gestational intermittent hypoxia; GNX, gestational normoxia; S1, stimulus rat 1; S2, stimulus rat 2.

parameters in addition to time spent in central versus peripheral areas of the field. No differences in movement speed, activity time, distance traveled, or tendency to explore the outer, middle, or center zones were detected in male GIH versus male GNX offspring (**Fig 3I–3L, S4A–S4G Fig**). Interestingly, however, we found that male GIH offspring spent more time grooming, exhibited an increase in the number of grooming episodes, and had a decreased latency to initiate grooming than male GNX offspring (**Fig 3M–3O**). The specificity of these findings to grooming was indicated by a lack of comparable changes in rearing time or rearing bouts (**S4H and S4I Fig**). Next, we wanted to evaluate the possibility that cognitive and/or social impairments emerge in GIH female offspring during adulthood (**S5A Fig**). In contrast to adult male GIH offspring, in adult female GIH offspring, we detected no differences in spontaneous alternation (**S5B Fig**), novel object recognition (**S5C and S5D Fig**), social approach (**S5E and S5F Fig**), or preference for social novelty (**S5G Fig**) relative to age-matched female GNX offspring. Further, no differences in any open field behaviors, including grooming behavior, were detected in female GIH offspring (**S6A–S6Q Fig**). Collectively, our findings indicate that cognitive and social deficits in male GIH offspring persist into adulthood, while adult female GIH offspring do not show evidence of cognitive or social aberrations.

## Dendritic spine density and morphology is altered in GIH offspring

Dendritic spines are the sites of most excitatory connections in the central nervous system, and alterations in prefrontal cortical dendritic spine density and/or morphology are a common hallmark in human conditions and associated animal models typified by impaired cognitive and social behavior [66,67]. First, we assessed dendritic spine density and morphology along apical dendrites of layer 2/3 pyramidal neurons from the medial prefrontal cortex (mPFC), an area important for both cognitive and social behaviors [68–72]. Dendritic spines were illuminated using DiOlistic delivery of red fluorescent DiI to coronal brain sections from late adolescent (8 weeks old) GIH and GNX offspring (**Fig 4A**). In layer 2/3 mPFC pyramidal neurons, we found that 8-week-old male GIH offspring exhibited a robust approximately 70% increase in total dendritic spine density relative to male GNX offspring (**Fig 4B and 4C**). A numerically more modest (approximately 20%), yet statistically significant, increase in total spine density was also detected in 8-week-old female GIH offspring as compared to female GNX offspring (**Fig 4B–4D**). To determine whether the increase in total spine density in GIH offspring relative to GNX offspring was statistically greater in males versus females, we assessed the interaction between GNX/GIH status and sex using a 2-way ANOVA and found a significant interaction between these parameters [$F(1,12) = 8.203$, $p = 0.0142$].

Next, we evaluated the possibility that the increase in layer 2/3 total spine density in 8-week-old GIH offspring was due to an increase in the density of a particular spine morphological subtype. On the basis of spine head morphology, spines were semiautomatically classified into 3 subtypes—thin, stubby, or mushroom—using our established methods [73–75]. Spines lacking a discernable neck were classified as stubby. Because thin and mushroom spines, on the other hand, exhibit a clear neck, the distinction between these spine subtypes was made on the basis of the spine head diameter (see Method details); the head diameter of mushroom spines is by definition larger than that of thin spines. Layer 2/3 spine subtype analysis in 8-week-old male rats revealed a main effect for GNX/GIH status [$F(1,18) = 12.98$, $p = 0.0020$]. Relative to male GNX offspring, post hoc analysis identified a significant increase in the density of thin spines in male GIH offspring, with no corresponding alterations in the density of stubby or mushroom spines (**Fig 4E**). By contrast, no main effect of GNX/GIH status was detected following spine subtype analysis in female rats [$F(1,18) = 0.3413$, $p = 0.5663$],

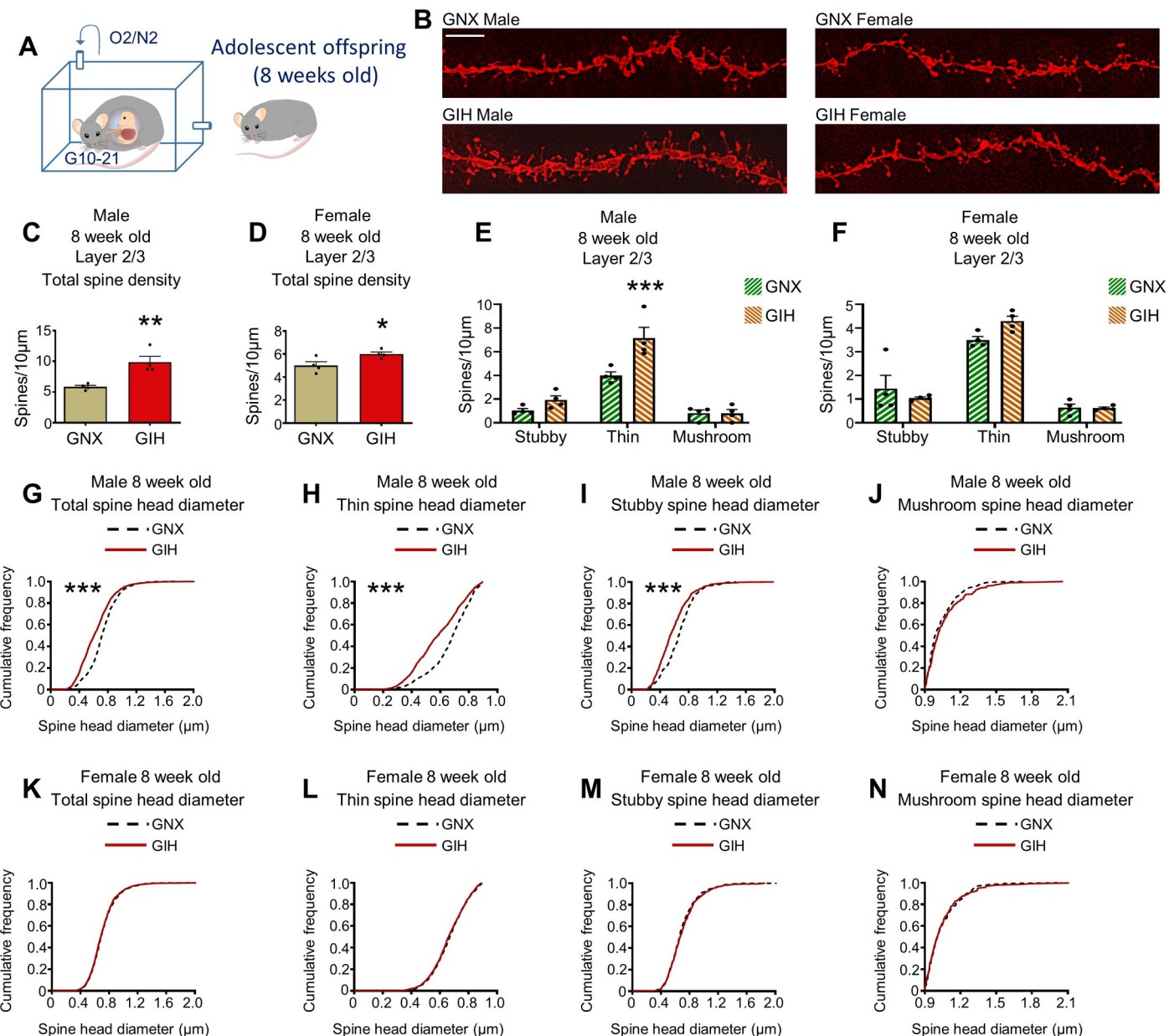

**Fig 4. mPFC pyramidal neuronal dendritic spine density and morphology is altered in layer 2/3 in 8-week-old GIH offspring. (A)** Schematic depicting GNX and GIH offspring age pertaining to all data in **Fig 4**. **(B)** Representative layer 2/3 pyramidal neuron dendrite segments from 8-week-old male and female GNX and GIH offspring. Scale bar = 10 μm. **(C)** Assessment of layer 2/3 total dendritic spine density revealed a significant increase in male GIH offspring relative to male GNX offspring [t (df,6) = 4.065, $p = 0.0066$]. $n = 4$ GNX and 4 GIH rats. **(D)** Assessment of layer 2/3 total dendritic spine density revealed a significant increase in female GIH offspring relative to male GNX offspring [t (df,6) = 2.651. $p = 0.0380$]. $n = 4$ GNX and 4 GIH rats. **(E)** Spine subtype density analysis revealed a significant increase in thin spine density in male GIH offspring [Bonferroni post hoc (df,18) $p = 0.0004$] and no alteration in the density of stubby [Bonferroni post hoc (df,18) $p = 0.5560$] or mushroom spines [Bonferroni post hoc (df,18) $p = 1.0$]. $n = 4$ GNX and 4 GIH rats. **(F)** No differences in the density of any spine subtypes were identified between female GIH and GNX offspring [Bonferroni post hoc stubby (df,18) $p = 0.8745$, thin $p = 0.1296$, mushroom $p = 1.0$]. $n = 4$ GNX and 4 GIH rats. **(G)** Analysis of cumulative total spine head diameter curves revealed a significant leftward shift in the curve for male GIH offspring (median survival differential 15.58%; $\chi^2 = 229.0$, $p < 0.0001$). $n = 1,750$ GNX and 2,869 GIH spines. **(H)** Analysis of cumulative thin spine head diameter curves revealed a significant leftward shift in the curve for male GIH offspring (median survival differential 15.24%; $\chi^2 = 168.3$, $p < 0.0001$). $n = 1,169$ GNX and 1,974 GIH spines. **(I)** Analysis of cumulative stubby spine head diameter curves revealed a significant leftward shift in the curve for male GIH offspring (median survival differential 18.41%; $\chi^2 = 34.67$, $p < 0.0001$). $n = 302$ GNX and 618 GIH spines. **(J)** No significant differences in the cumulative mushroom spine head diameter curve were identified between male GIH and GNX offspring (median survival differential 2.3%). $n = 279$ GNX and 277 GIH spines. **(K–N)** No significant differences in total or spine subtype cumulative head diameter curves were identified between female GIH and GNX offspring (total spine median survival differential 0.63%, thin differential 1.14%, stubby differential 1.12%, and mushroom differential 0.15%). $n = 2,682$ GNX and 2,245 GIH total spines; 1,866 GNX and 1,635 GIH thin spines; 507 GNX and 374 GIH stubby spines; 309 GNX and 236 GIH mushroom spines. All bar graphs are the mean + SEM. The data underlying this figure can be found in S1 Raw Data. GIH, gestational intermittent hypoxia; GNX, gestational normoxia; mPFC, medial prefrontal cortex.

and post hoc analysis failed to identify alterations in the density of any spine subtypes in female GIH versus GNX offspring (**Fig 4F**).

As a further assessment of spine morphology, we analyzed cumulative frequencies of spine head diameters across all spines and within individual spine subtypes. In 8-week-old male GIH offspring, we detected a significant leftward shift in the cumulative layer 2/3 total spine head diameter curve relative to male GNX offspring (**Fig 4G**), indicating an increased accumulation of spines with smaller head sizes in male GIH offspring. Further, in male GIH offspring, we detected a significant leftward shift in the curve for both thin and stubby spine subclasses, with no curve shift for mushroom spines (**Fig 4H–4J**). On the other hand, no differences in the total or spine subtype cumulative head diameter curves were detected between female GNX and GIH offspring (**Fig 4K–4N**). Collectively, these findings indicate that in male GIH offspring the leftward shift in the total spine head diameter curve is the likely result of an increase in the density of thin spines (which have small heads) in addition to a decrease in the head diameter of thin and stubby spines.

We also analyzed dendritic spines along apical dendrites in layer 5 mPFC pyramidal neurons from 8-week-old GIH and GNX offspring (**Fig 5A**). Layer 5 mPFC neuron analysis identified a robust and significant up-regulation (52% increase) of total spine density in male GIH offspring relative to male GNX offspring (**Fig 5B and 5C**). On the other hand, no significant alteration in total spine density in layer 5 neurons was detected in female GIH offspring (**Fig 5B–5D**). Spine subtype analysis in male GIH offspring identified a main effect for maternal GIH/GNX status [$F(1,18) = 6.839$, $p = 0.0175$]. Post hoc analysis revealed that that the increase in total spine density in male GIH offspring is the product of a significant increase in thin spines, with no differences in stubby or mushroom spines (**Fig 5E**). By contrast, no main effect of maternal GIH/GNX status was identified in female offspring [$F(1,18) = 2.437$, $p = 0.1359$], and no post hoc differences were detected in any spine subtypes (**Fig 5F**).

Cumulative frequency assessments also identified male-specific alterations in layer 5 mPFC spine head diameters in 8-week-old GIH offspring. Specifically, we detected a significant leftward shift in the total spine head diameter curve in male GIH offspring relative to male GNX offspring (**Fig 5G**) indicative of a reduction in collective spine head diameter in male GIH offspring. Among spine subtypes, we detected a significant leftward shift in thin and stubby head diameter curves in male GIH offspring, with no alteration in mushroom spine curves (**Fig 5H–5J**). No corresponding differences in cumulative spine head diameter curves were detected in female GIH offspring, however (**Fig 5K–5N**). These data indicate that the reduction in male GIH total spine head diameter in layer 5 neurons is the result of an increase in thin spine numbers and a decrease in the head diameter of thin and stubby spines.

Starting at birth and continuing into early childhood, a period of increased synaptogenesis is responsible for the surge in dendritic spine numbers in cortical pyramidal neurons. Starting around midchildhood and extending into adolescence, cortical spine density decreases, owing to an increase in the pruning of superfluous spines. By late adolescence/young adulthood, cortical spine formation and elimination are balanced such that net spine density is relatively constant throughout most of adulthood [66,76]. It is possible that the increase in mPFC spine density in 8-week-old male GIH offspring is the product of an attenuation of normal synaptic pruning events, the result of an increase in synaptogenesis during early childhood, or a combination of both processes. To distinguish between these possibilities, we evaluated spine density in layer 2/3 and layer 5 mPFC neurons from 3-week-old male GIH and GNX offspring (**Fig 6A**). In 3-week-old GIH offspring, we found that total spine density was significantly increased in mPFC neurons from layer 2/3, but not layer 5, as compared to GNX offspring (**Fig 6B–6D**). Layer 2/3 spine subtype analysis revealed a main effect of maternal GNX/GIH status [$F(1,18) = 7.865$, $p = 0.0117$], with post hoc tests identifying a specific increase in thin spine density only

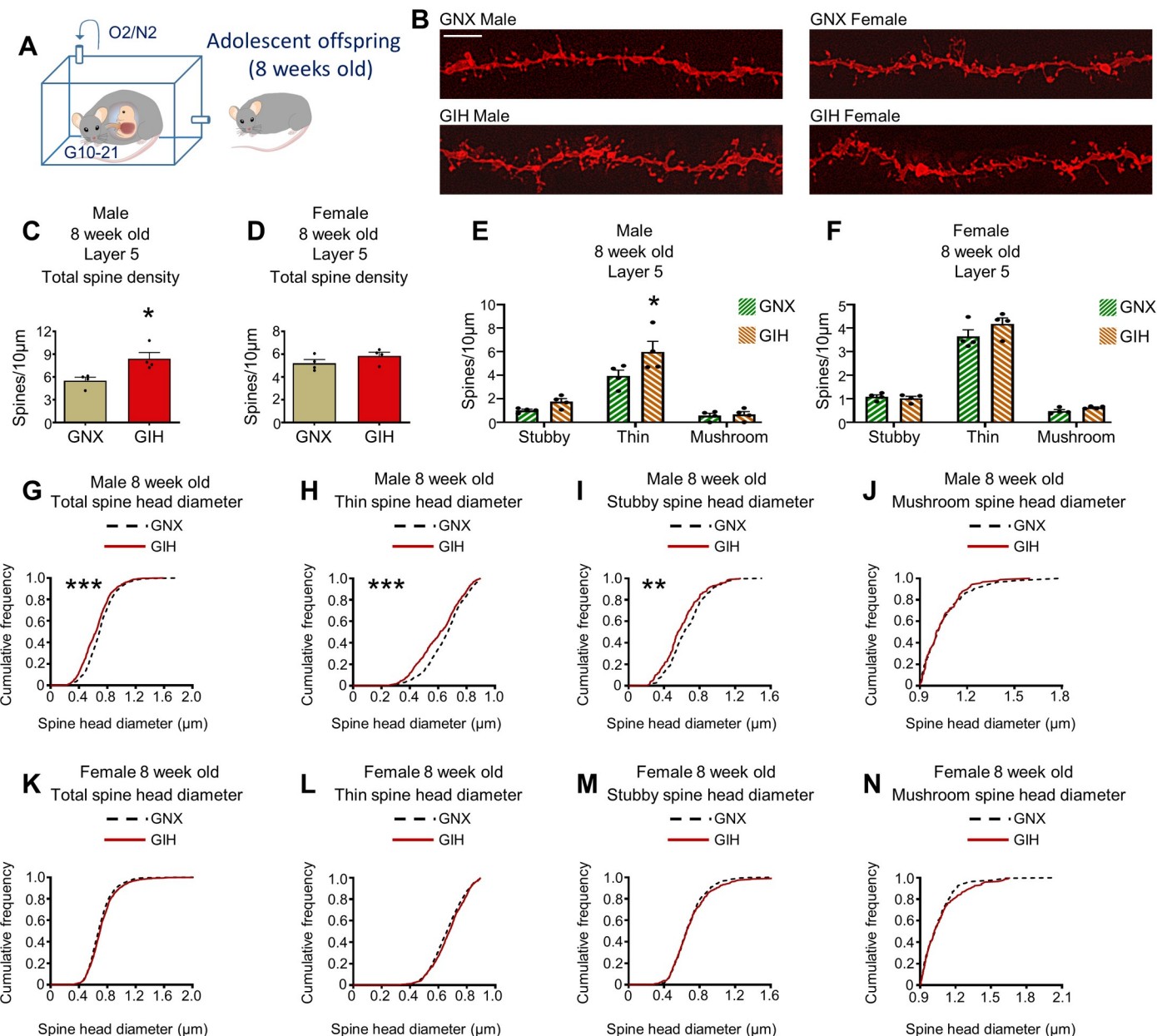

**Fig 5. mPFC pyramidal neuronal dendritic spine density and morphology is altered in layer 5 in 8-week-old GIH offspring. (A)** Schematic depicting GNX and GIH offspring age pertaining to all data in Fig 5. **(B)** Representative layer 5 pyramidal neuron dendrite segments from 8-week-old male and female GNX and GIH offspring. Scale bar = 10 μm. **(C)** Assessment of total layer 5 dendritic spine density revealed a significant increase in male GIH offspring relative to male GNX offspring [t (df,6) = 3.109, $p$ = 0.0209]. $n$ = 4 GNX and 4 GIH rats. **(D)** Assessment of total layer 5 dendritic spine density reveals no significant differences between female GIH and GNX offspring [t (df,6) = 1.391, $p$ = 0.2136]. $n$ = 4 GNX and 4 GIH rats. **(E)** Spine subtype density analysis reveals a significant increase in thin spine density in male GIH offspring [Bonferroni post hoc (df,18) $p$ = 0.0145] with no alteration in the density of stubby [Bonferroni post hoc (df,18) $p$ = 0.7802] or mushroom spines [Bonferroni post hoc (df,18) $p$ = 1.0]. $n$ = 4 GNX and 4 GIH rats. **(F)** No differences in the density of any spine subtypes were identified between female GIH and GNX offspring [Bonferroni post hoc (df,18) stubby $p$ = 1.0; thin $p$ = 0.0987; mushroom $p$ = 1.0]. $n$ = 4 GNX and 4 GIH rats. **(G)** Analysis of cumulative total spine head diameter curves revealed a significant leftward shift in the curve for male GIH offspring (median survival differential 7.4%; $\chi^2$ = 53.36, $p$ < 0.0001). $n$ = 1,702 GNX and 979 GIH spines. **(H)** Analysis of cumulative thin spine head diameter curves revealed a significant leftward shift in the curve for male GIH offspring (median survival differential 8.0%; $\chi^2$ = 37.64, $p$ < 0.0001). $n$ = 1,180 GNX and 685 GIH spines. **(I)** Analysis of cumulative stubby spine head diameter curves revealed a significant leftward shift in the curve for male GIH offspring (median survival differential 10.52%; $\chi^2$ = 9.605, $p$ = 0.0019). $n$ = 310 GNX and 204 GIH spines. **(J)** No significant differences in the cumulative mushroom spine head diameter curve were identified between male GIH and GNX offspring (median survival differential 0.78%). $n$ = 212 GNX and 90 GIH spines. **(K–N)** No significant differences in total or spine subtype cumulative head diameter curves were identified between female GIH and GNX offspring (total spine median survival differential 3.23%, thin differential 2.6%, stubby differential 0.78%, and mushroom differential 0.4%). $n$ = 2,376 GNX and 1,682 GIH total spines; 1,683 GNX and 1,155 GIH thin spines; 469 GNX and 323 GIH stubby spines; 224 GNX and 204 GIH mushroom spines. All bar graphs are the mean + SEM. The data underlying this figure can be found in S1 Raw Data. GIH, gestational intermittent hypoxia; GNX, gestational normoxia; mPFC, medial prefrontal cortex.

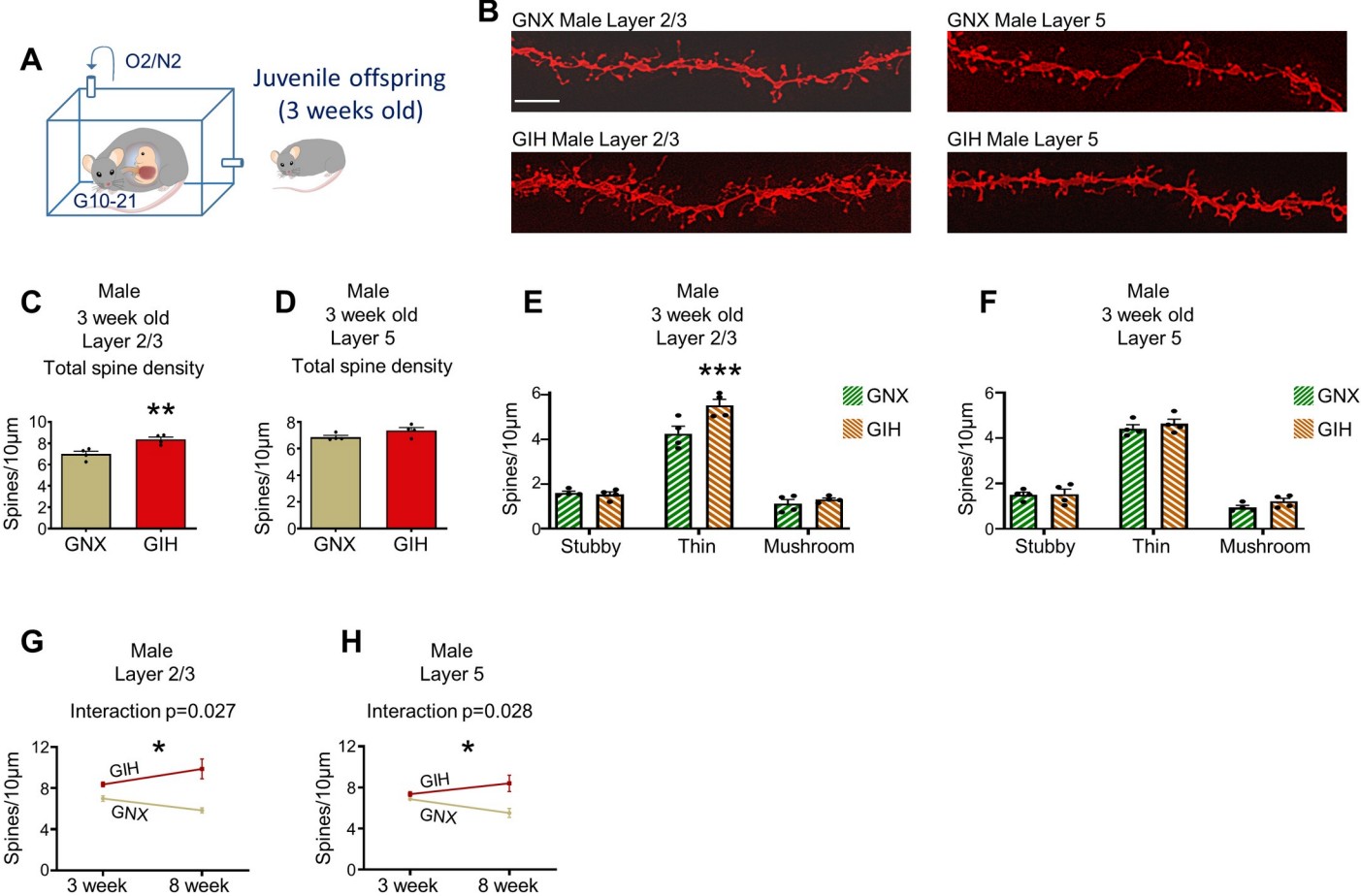

**Fig 6. mPFC pyramidal neuronal dendritic spine density and morphology in layer 2/3 and layer 5 in 3-week-old male GIH offspring. (A)** Schematic depicting GNX and GIH offspring age pertaining to all data in Fig 6. **(B)** Representative layer 2/3 and layer 5 pyramidal neuron dendrite segments from 3-week-old male GNX and GIH offspring. Scale bar = 10 μm. **(C)** Assessment of total layer 2/3 dendritic spine density revealed a significant increase in male GIH offspring relative to male GNX offspring [t (df,6) = 3.888, $p$ = 0.0081]. $n$ = 4 GNX and 4 GIH rats. **(D)** Assessment of total layer 5 dendritic spine density revealed no differences in male GIH offspring relative to male GNX offspring [t (df,6) = 1.884, $p$ = 0.1086]. $n$ = 4 GNX and 4 GIH rats. **(E)** Spine subtype density analysis revealed a significant increase in layer 2/3 thin spine density in male GIH offspring [Bonferroni post hoc (df,18) $p$ = 0.0009] and no alteration in the density of stubby [Bonferonni post hoc (df,18) $p$ = 1.0] or mushroom spines [Bonferroni post hoc (df,18) $p$ = 1.0]. $n$ = 4 GNX and 4 GIH rats. **(F)** No differences in the density of any layer 5 spine subtypes were identified between male GIH and GNX offspring [Bonferroni post hoc (df,18) stubby $p$ = 1.0; thin $p$ = 1.0; mushroom $p$ = 0.7706]. $n$ = 4 GNX and 4 GIH rats. **(G)** A significant interaction was detected between layer 2/3 total spine density in GIH versus GNX offspring as a function of age [2-way ANOVA F(1,12) = 6.366, $p$ = 0.0268]. **(H)** A significant interaction was detected between layer 5 total spine density in GIH versus GNX offspring as a function of age [2-way ANOVA F(1,12) = 6.293, $p$ = 0.0275]. All bar graphs are the mean + SEM. The data underlying this figure can be found in S1 Raw Data. GIH, gestational intermittent hypoxia; GNX, gestational normoxia; mPFC, medial prefrontal cortex.

(**Fig 6E**). By contrast, layer 5 subtype analysis in 3-week-old animals showed no main effect for maternal GNX/GIH status [F(1,18) = 1.601, $p$ = 0.2219] and no differences in spine subtype density between male GNX and GIH conditions (**Fig 6F**). Finally, spine head cumulative frequency analysis revealed no significant alterations in the curve between 3-week-old GNX and GIH offspring across all spine subtypes or within individual spine subtypes for layers 2/3 and 5 mPFC neurons (**S7A–S7I Fig**).

We noted that relative to age matched GNX offspring, the increased spine density in male GIH offspring is numerically more modest at 3 weeks of age compared to 8 weeks. A statistical interaction between GNX/GIH status and age in both layer 2/3 [F(1,12) = 6.366, $p$ = 0.0268] and layer 5 neurons [F(1,12) = 6.293, $p$ = 0.0275] confirmed the increased magnitude of spine

density gains with age in male GIH offspring relative to GNX offspring (**Fig 6G and 6H**). Collectively, our findings suggest that elevations in mPFC pyramidal neuron spine density in male GIH offspring likely arise largely from a decrease in spine pruning spanning childhood to adolescence and, in the case of layer 2/3 neurons, also due to a slight increase in spine formation during early postnatal development.

To determine if robust increased spine density among 8-week-old male GIH offspring are characteristic of pyramidal neurons in other forebrain regions, we assessed spine density and morphology in hippocampal CA3 field pyramidal neurons from 8-week-old rats (**S8A Fig**). In contrast to what we observed for the mPFC, we did not detect a significant difference in total spine density in CA3 neurons in male GIH offspring compared to GNX offspring (**S8B and S8C Fig**). Nevertheless, while spine subtype analysis did not identify a main effect for GNX/GIH status [$F(1,12) = 2.985$ $p = 0.1097$], post hoc analysis indicates that GIH offspring show a significant 15% increase in the density of thin spines, with no corresponding alterations in stubby or mushroom spine numbers (**S8D Fig**). Further, no differences in CA3 cumulative spine head diameter curves were identified between GNX and GIH offspring (**S8E–S8H Fig**).

## AMPAergic function is altered in GIH offspring

Next, we wanted to determine if the increase in mPFC dendritic spine density in male GIH offspring is associated with alterations in synaptic function. To this end, we assessed AMPA-mediated spontaneous excitatory postsynaptic currents (sEPSCs) in layer 2/3 mPFC pyramidal neurons in adult male GIH and GNX offspring. Firing patterns in response to current injection were used to identify putative pyramidal neurons (**Fig 7A**) and sEPSC interevent interval (inverse of frequency), amplitude, rise time, and half width were quantified. Voltage clamp traces for GNX and GIH offspring are shown in **Fig 7B**, and detected sEPSCs are shown in **Fig 7C**. For each parameter, 2 modes of analysis were performed. First, pairwise assessments were made between each GNX and GIH offspring neuron using a comparison matrix in which $p$-values were calculated using Kruskal–Wallis followed by Bonferroni–Holm corrections for multiple comparisons. Second, a direct statistical group-wise comparison of each parameter between GNX and GIH offspring was also made. For sEPSC interevent intervals, we found that randomly chosen neurons from GNX and GIH offspring have significantly different intervals 98% of the time (41/42 pairwise comparisons), and in these instances, GNX neurons were 2.1 times (28/13) more likely to have the greater interval and hence lower sEPSC frequency (**Fig 7D and 7E**). Further, direct comparison revealed that GIH offspring neurons had a significantly greater sEPSC frequency compared to GNX offspring neurons (**Fig 7F**). For sEPSC amplitude, we found that randomly chosen neurons from GNX and GIH offspring had significantly different amplitudes 86% of the time (36/42 pair wise comparisons), and in these instances, GNX neurons were 3 times (27/9) more likely to have the larger amplitude (**Fig 7G and 7H**). However, direct group comparisons between GNX and GIH offspring did not quite reach statistical significance ($p = 0.06$) (**Fig 7I**). By contrast, analysis of sEPSC rise time and half widths did not identify clear group differences between GNX and GIH offspring (**Fig 7J–7O**).

## Aberrant mTOR signaling and rescue of behavioral deficits in GIH male offspring

Next, we wanted to begin to identify biochemical processes that are potentially involved in contributing to key behavioral phenotypes in GIH male offspring. To this end, we microdissected mPFC tissue from 3-week-old (juvenile) GNX and GIH offspring rats (**Fig 8A**), an age in which spine changes in the GIH group are minimal, and thus any identified biochemical

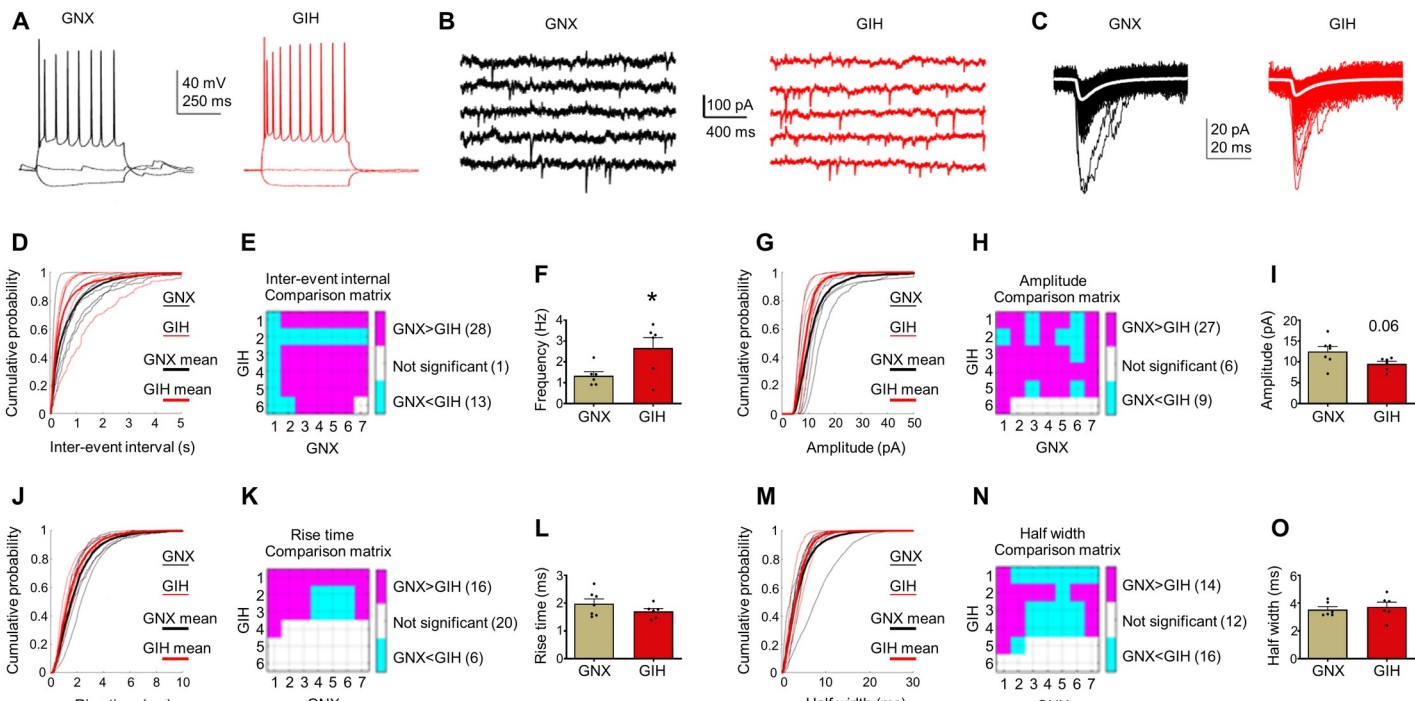

**Fig 7. Alterations in AMPA sEPSCs from mPFC layer 2/3 pyramidal neurons of adult male GIH offspring. (A)** Firing patterns in response to current injections (−100, 0, 280 pA) were used to identify putative pyramidal neurons in GNX and GIH offspring. **(B)** Examples of continuous voltage clamp traces (−60 mV, room temperature) from layer 2/3 mPFC pyramidal neurons in GNX and GIH adult offspring. **(C)** Detected sEPSCs (black or red) and average (white) from the same cells as in B. **(D)** Cumulative distributions of sEPSC interevent intervals in adult GNX and GIH offspring. **(E)** Interevent interval comparison matrix shows differences and their directions for all pairwise comparisons between adult GNX and GIH offspring neurons. **(F)** Graph shows mean group sEPSC frequency for GNX and GIH offspring neurons [t (df,10) = 2.473, p = 0.033]. n = 6 GNX and 6 GIH cells from 2 rats each (1 Grubbs test extreme outlier in GNX group removed). **(G)** Cumulative distributions of sEPSC amplitudes in adult GNX and GIH offspring. **(H)** Amplitude comparison matrix shows differences and their directions for all pairwise comparisons between GNX and GIH offspring neurons. **(I)** Graph shows mean group sEPSC amplitude for adult GNX and GIH offspring neurons [t (df,11) = 2.052, p = 0.0647]. n = 7 GNX and 6 GIH cells from 2 rats each. **(J)** Cumulative distributions of sEPSC rise times in adult GNX and GIH offspring. **(K)** Rise time comparison matrix shows differences and their directions for all pairwise comparisons between GNX and GIH offspring neurons. **(L)** Mean group sEPSC rise time for adult GNX and GIH offspring neurons [t (df,11) = 1.344, p = 0.2059]. n = 7 GNX and 6 GIH cells from 2 rats each. **(M)** Cumulative distributions of sEPSC half widths in GNX and GIH offspring. **(N)** Half width comparison matrix shows differences and their directions for all pairwise comparisons between GNX and GIH offspring neurons. **(O)** Graph shows mean group sEPSC half widths for adult GNX and GIH offspring neuron [t (df,10) = 0.4700, p = 0.6484]. n = 6 GNX and 6 GIH cells from 2 rats each (1 Grubbs test extreme outlier in GNX group removed—same neuron that was outlier for frequency). All bar graphs are the mean + SEM. The data underlying this figure can be found in S1 Raw Data. GIH, gestational intermittent hypoxia; GNX, gestational normoxia; mPFC, medial prefrontal cortex; sEPSC, spontaneous excitatory postsynaptic current.

alterations are less likely to be mere consequences of altered synaptic connectivity than if we investigated adolescent rats. mPFC homogenates were resolved via SDS-PAGE followed by western blotting. We focused on the mTOR pathway as hyperactive mTOR signaling has been identified in the cortex of autism subjects [77]. Further, many monogenic alterations that greatly increase risk for the development of autism adversely impact the integrity of genes whose protein products normally act as brakes for the mTOR pathway, thereby resulting in excessive mTOR activity [78,79]. Finally, dampening mTOR activity has been postulated as a potential therapeutic approach for autism spectrum disorders [80].

mTOR exists in 2 distinct protein complexes, mTORC1 and mTORC2, in which mTORC1 is mTOR in a complex with the Raptor protein, while mTORC2 is mTOR in a complex with the Rictor protein [81–83]. mTORC1 and mTORC2 have many divergent functions, and notably, mTORC1 has been implicated in increasing synapse stability in cortical neurons by reducing the autophagy of dendritic spines and by increasing the local translation of a select groups of mRNAs present in dendritic spines [84,85]. The phosphorylation of mTOR at serine residue

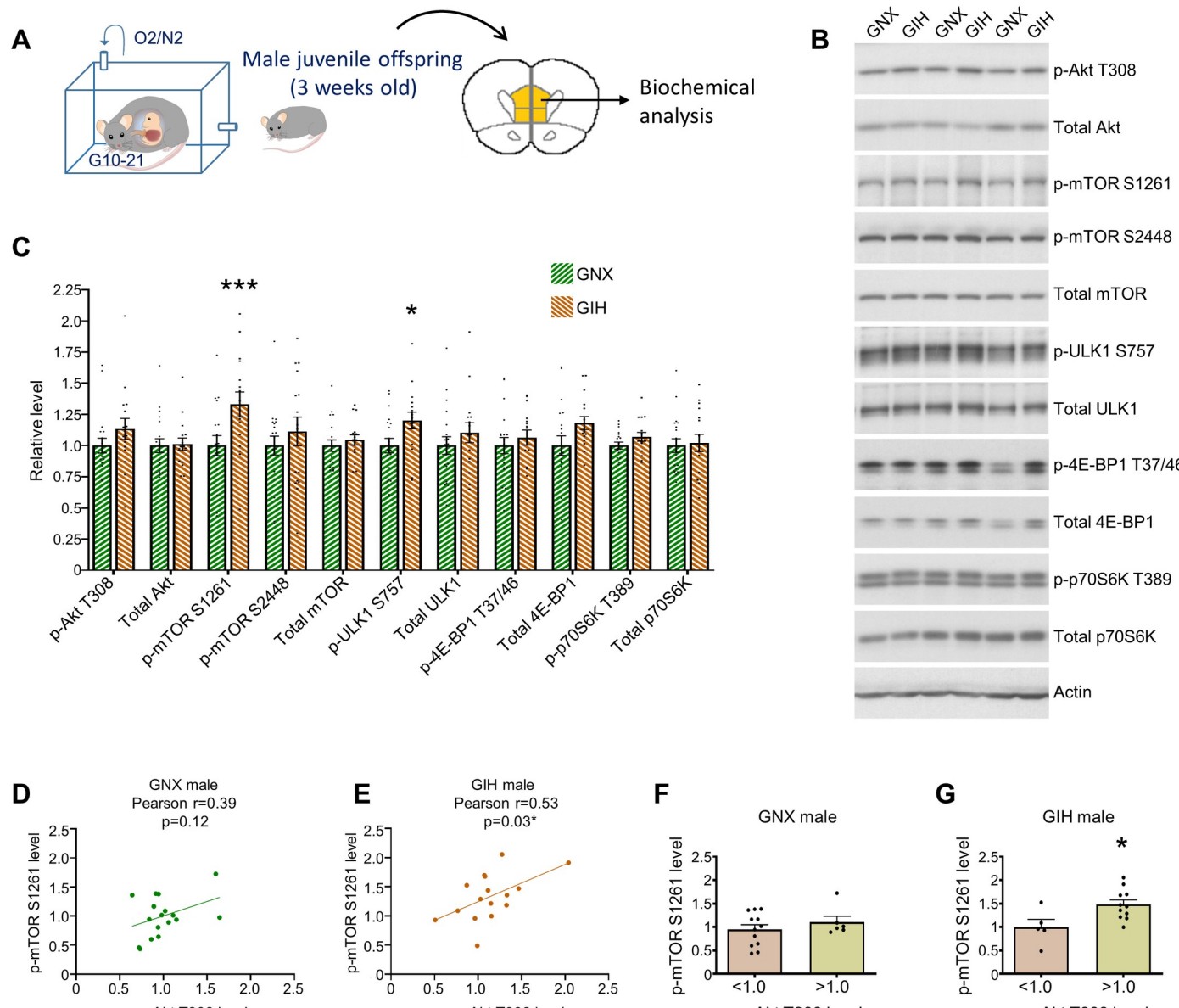

**Fig 8. Biochemical analysis of mPFC homogenates in 3-week-old male offspring. (A)** Schematic depicting age of GNX and GIH male offspring used for biochemical analysis of mPFC homogenates. **(B)** Representative western blots showing indicated phospho or total proteins in 3-week-old GNX and GIH male offspring mPFC homogenates. **(C)** Quantification of indicated phospho and total protein levels in 3-week-old GNX and GIH male offspring mPFC homogenates (normalized to actin). One-way ANOVA with Fisher LSD: p-Akt 308 ($p = 0.1614$), total Akt ($p = 0.9063$), p-mTOR S1261 (***$p = 0.0007$), p-mTOR S2448 ($p = 0.2364$), total mTOR ($p = 0.6305$), p-ULK1 S757 (*$p = 0.0354$), total ULK1 ($p = 0.2784$), p-4E-BP1 T37/46 ($p = 0.4951$), total 4E-BP1 ($p = 0.0586$), p-p70S6K T389 ($p = 0.4579$), total p70S6K ($p = 0.8239$). $n = 19$ GNX and 16 GIH. **(D)** Correlation analysis performed in mPFC homogenates from 3-week-old GNX male offspring. p-Akt T308 levels did not significantly correlate with levels of p-mTOR S1261. Statistics indicated above graph. **(E)** Correlation analysis performed in mPFC homogenates from 3-week-old GIH male offspring. p-Akt T308 levels significantly correlated with levels of p-mTOR S1261. Statistics indicated above graph. **(F)** Binning of p-Akt T308 levels in 3-week-old GNX male offspring homogenates. Average GNX offspring p-Akt T308 levels are 1.0, and data split into 2 groups— samples with p-Akt T308 levels below the mean level for GNX offspring ($<1.0$) and samples with p-Akt T308 levels above the mean level for GNX offspring ($>1.0$). Levels of p-mTOR S1261 in the below and above average p-Akt T308 groups are shown. Above average p-Akt T308 levels did not result in greater levels of p-mTOR S1261 compared to samples with below average p-Akt T308 levels [t(16) = 0.9259; $p = 0.3682$]. **(G)** Binning of p-Akt T308 levels in 3-week-old GIH male offspring homogenates. GIH offspring p-Akt T308 levels were binned into 2 groups as described in (F)—those below the 1.0 mean for GNX ($<1.0$) and those above the 1.0 mean for GNX ($>1.0$). GIH offspring samples with p-Akt T308 levels greater than the mean of GNX offspring have significantly higher levels of p-mTOR S1261 than GIH offspring samples with p-Akt T308 levels below the mean for GNX offspring [t(14) = 2.646; $p = 0.0192$]. All bar graphs are the mean + SEM. The data underlying this figure can be found in S1 Raw Data. GIH, gestational intermittent hypoxia; GNX, gestational normoxia; LSD, least significant difference; mPFC, medial prefrontal cortex; mTOR, mammalian target of rapamycin.

1261 (S1261) is a consequence of feedforward activity from kinases and other proteins operating upstream of mTOR, and increase phosphorylation of this residue facilitates mTORC1 activity [86]. We found that levels of mTOR phosphorylated at S1261 are significantly increased in the mPFC of juvenile GIH rats relative to GNX rats (**Fig 8B and 8C**).

Interestingly, in GIH male offspring, we did not find a significant alteration in the activity of Akt (**Fig 8B and 8C**), a kinase that is known to indirectly facilitate mTOR activity via a series of intermediate proteins, as evidenced by a lack of significant change in phosphorylation at the threonine 308 residue (T308), which augments Akt activity [82]. Despite this, our findings suggest that the relationship between mPFC Akt activity and mTOR activity is augmented in GIH male offspring relative to GNX male offspring, as levels of Akt T308 phosphorylation were moderately and nonsignificantly correlated with mTOR S1261 phosphorylation in GNX mPFC homogenates (**Fig 8D**), while exhibiting a stronger, statistically significant correlation in GIH mPFC homogenates (**Fig 8E**). To further examine this relationship, we binned Akt T308 phosphorylation levels in individual GNX and GIH mPFC samples into those above the mean level for GNX (>1.0) and those below the mean level for GNX (<1.0). We found that GNX homogenates with above average Akt T308 phosphorylation (>1.0) did not have higher levels of mTOR S1261 phosphorylation relative to GNX homogenates with below average Akt T308 phosphorylation levels (<1.0) (**Fig 8F**). On the other hand, in GIH mPFC homogenates, above average Akt T308 phosphorylation was associated with higher levels of mTOR S1261 phosphorylation compared to GIH homogenates with below average levels of Akt T308 phosphorylation (**Fig 8G**). Taken together, these findings indicate that despite the lack of significant global changes in phospho-Akt T308, Akt phosphorylation is more predictive and perhaps a stronger driver of mTOR phospho-activity in GIH mPFC homogenates as compared to GNX.

Next, we examined the activity of known mTORC1 downstream targets, including ULK1, a kinase known to promote autophagy, and 4E-BP1 and p70S6K, which play distinct roles in the regulation of protein translation [87–89]. mTOR phosphorylates ULK1 at the serine 757 residue (S757), thereby inhibiting ULK1 activity with a consequent inhibition of autophagy [88,90]. We found that ULK1 S757 phosphorylation was significantly increased in juvenile mPFC GIH offspring homogenates compared to GNX, with no changes in total ULK1 levels detected (**Fig 8B and 8C**). On the other hand, known mTORC1 phosphorylation sites on 4E-BP1 (threonine 37/46; T37/46) and p70S6K (threonine 389; T389) [91–93] were not altered in GIH homogenates, and total levels of these proteins were also not altered (**Fig 8B and 8C**). While mTOR phosphorylation at S1261 is a consequence of feedforward mTOR pathway activity [86], mTOR phosphorylation at the serine 2448 residue (S2448) is a product of a feedback signaling loop in which mTOR-mediated activation of p70S6K results in the direct phosphorylation mTOR at S2448 via p70S6K, thereby providing additional mTOR activation [94,95]. Consistent with the lack of change in levels of phospho-active p70S6K in GIH offspring homogenates, we found no changes in mTOR S2448 phosphorylation in juvenile mPFC GIH offspring homogenates compared to GNX (**Fig 8B and 8C**). Collectively, these findings are suggestive of a level of specificity toward increased mTORC1-mediated phosphoinhibition of the pro-autophagy kinase ULK1 in the mPFC of GIH offspring.

To determine if dampening mTOR activity in GIH male offspring is able to alleviate deficits in cognition and social behavior, we implanted diffusion pellets containing rapamycin, a specific and well-established inhibitor of the mTORC1 complex [96], into the midscapular region of 3-week-old juvenile GNX and GIH male rats (**Fig 9A**). The implanted pellets were designed to deliver rapamycin for 3 weeks. The 3-week delivery mechanism at the formulated concentration of rapamycin we used was based on previous work [97]. Using this approach, rapamycin delivery will cease when the rats are 6 weeks of age. Vehicle pellets were used as a control.

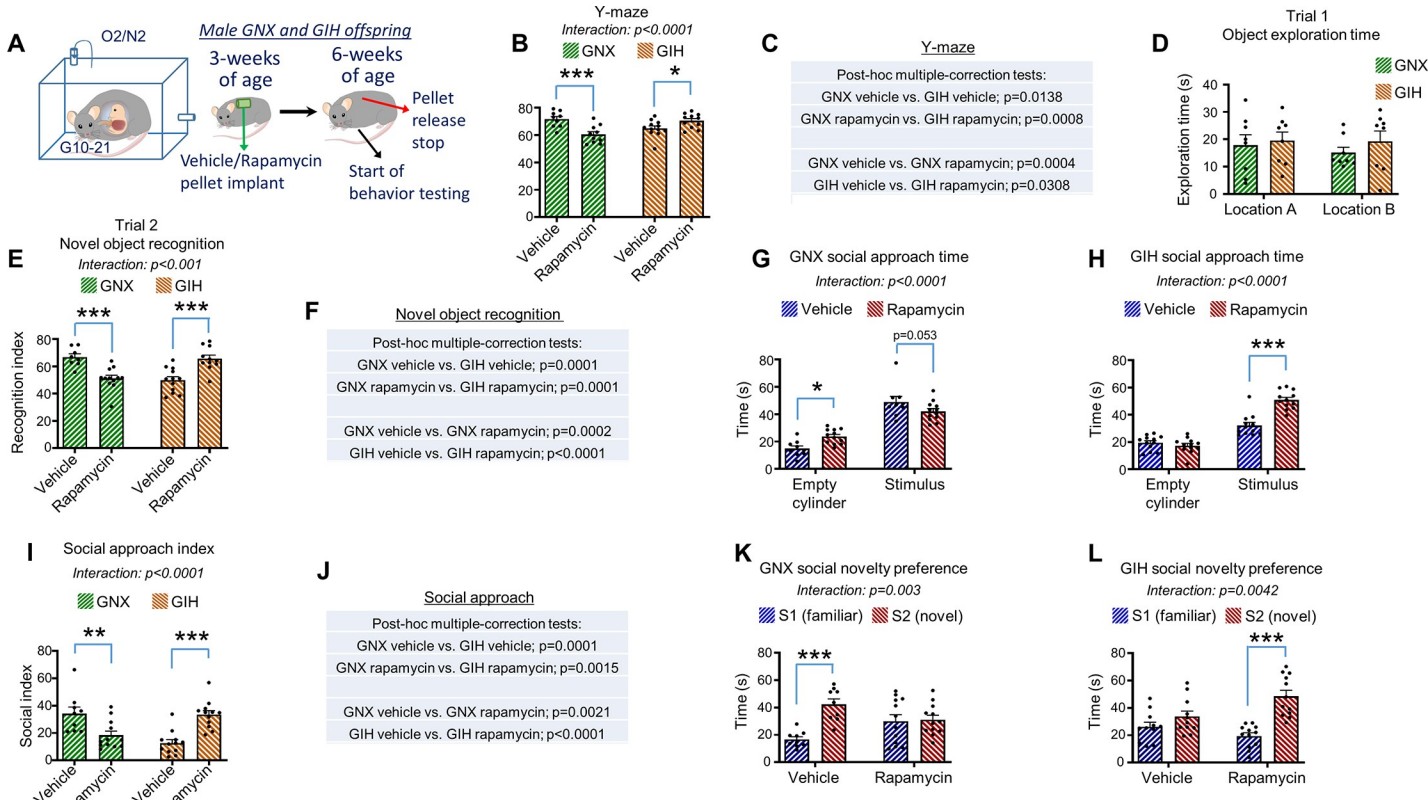

**Fig 9. Rapamycin rescue of GIH male behavioral impairments. (A)** Experimental approach. Three-week-old GNX and GIH male offspring were implanted with 21-day rapamycin or vehicle slow-release pellets. Following the cessation of the rapamycin or vehicle release (6 weeks of age), behavioral assessment commenced. **(B and C)** Y-maze spontaneous alternation. Relative to vehicle treatment, rapamycin significantly reduced alternation percentage in male GNX offspring and significantly increased alternation percentage in male GIH offspring. Statistical results are shown in the chart (2-way ANOVA with Holm–Sidak post hoc). $n$ = 9 GNX vehicle, 10 GNX rapamycin, 12 GIH vehicle, 11 GIH rapamycin. **(D)** Time spent exploring 2 identical objects in different locations during trial 1 of the novel object recognition test. No effects of GNX/GIH maternal status or vehicle/rapamycin treatment were detected. **(E and F)** Novel object recognition test trial (trial 2). Rapamycin significantly reduced the recognition index in GNX offspring relative to vehicle-treated GNX offspring and significantly increased the recognition index in GIH male offspring compared to vehicle-treated GIH offspring. Statistical results are shown in the adjacent chart (2-way ANOVA with Holm–Sidak post hoc). $n$ = 8 GNX vehicle, 12 GNX rapamycin, 12 GIH vehicle, 11 GIH rapamycin. **(G)** Social approach time in vehicle and rapamycin-treated GNX male offspring rats. Relative to the vehicle group, rapamycin increased the time spent investigating the empty cylinder [2-way ANOVA with Holm–Sidak post hoc (df,38) $p$ = 0.0292], with a trend toward decreasing time spent investigating the stimulus rat containing cylinder [2-way ANOVA with Holm–Sidak post hoc (df,38) $p$ = 0.0528]. $n$ = 9 vehicle, 12 rapamycin. **(H)** Social approach time in vehicle and rapamycin-treated GIH offspring rats. Relative to the vehicle group, rapamycin increased the time spent investigating the stimulus rat containing cylinder [2-way ANOVA with Holm–Sidak post hoc (df,44) $p$ < 0.0001]. $n$ = 12 vehicle, 12 rapamycin. **(I and J)** Social index score in the social approach test. Rapamycin reduced the social index score in GNX male rats and increased the social index score in GIH male rats. Statistical results are shown in the adjacent chart. $n$ = 9 GNX vehicle, 12 GNX rapamycin, 12 GIH vehicle, 12 GIH rapamycin. **(K)** Time spent investigating each rat in the preference for social novelty test in vehicle or rapamycin-treated male GNX offspring. Whereas vehicle GNX offspring showed a significant preference for the novel rats versus the familiar rat [2-way ANOVA with Holm–Sidak post hoc (df,38) $p$ = 0.0002], rapamycin-treated GNX offspring did not show a preference for the novel rat [2-way ANOVA with Holm–Sidak post hoc (df,38) $p$ = 0.8283]. **(L)** Time spent investigating each rat in the preference for social novelty test in vehicle or rapamycin-treated male GIH offspring. Vehicle GIH offspring did not show significant preference for the novel rat versus the familiar rat [2-way ANOVA with Holm–Sidak post hoc (df,42) $p$ = 0.1502]. However, rapamycin-treated GIH spent more time investigating the novel versus familiar rat [2-way ANOVA with Holm–Sidak post hoc (df,42) $p$ < 0.0001]. All bar graphs are the mean + SEM. The data underlying this figure can be found in S1 Raw Data. GIH, gestational intermittent hypoxia; GNX, gestational normoxia.

Rats were tested for spontaneous alternation, object recognition, and social behaviors in succession starting at 6 weeks of age with behavioral testing ending at 7.4 weeks of age. For Y-maze spontaneous alternation, we found that rapamycin had opposite effects on performance in GNX and GIH offspring. More specifically, while vehicle-treated GIH offspring rats performed significantly more poorly than vehicle-treated GNX offspring, rapamycin significantly improved performance in GIH offspring and significantly impaired performance in GNX offspring relative to their vehicle-treated counterparts [2-way ANOVA interaction: F(1,38) = 20.86, $p$ < 0.0001]

(**Fig 9B and 9C**). Pertaining to novel object recognition testing, while rapamycin did not affect the motivation to investigate objects during trial 1 (**Fig 9D**), rapamycin had opposing effects on novel object recognition in GNX versus GIH offspring during trial 2 [Interaction: $F_{(1,39)}$ = 37.57, $p < 0.0001$]. Indeed, vehicle-treated GIH offspring showed a significantly reduced preference for the novel object compared to vehicle-treated GNX offspring, with rapamycin improving novel object preference in the GIH offspring, yet impairing performance in the GNX offspring (**Fig 9E and 9F**).

Finally, rapamycin also differentially affected social behavior in adolescent male offspring of GNX and GIH mothers. For social approach, when investigating the total time GNX offspring spend investigating the empty cylinder versus the cylinder containing a stimulus rat, we found a significant interaction for rapamycin treatment and end chamber cylinder preference [$F_{(1,19)}$ = 8.392, $p$ = 0.0092], with post hoc testing indicating that the GNX rapamycin group spent more time investigating the empty cylinder than the GNX vehicle group, with a trend toward a decrease in time spent investigating the stimulus rat cylinder (**Fig 9G**). In GIH male offspring, we also detected a significant interaction for rapamycin treatment and end chamber cylinder [$F_{(1,22)}$ = 29.58, $p < 0.0001$], but opposite to GNX, we detected a significant increase in preference for the stimulus chamber in the GIH rapamycin group versus the GIH vehicle group (**Fig 9H**). To more directly compare the effects of rapamycin on social approach in GNX and GIH offspring, we examined social index scores. We detected a significant interaction for GNX/GIH offspring status and rapamycin treatment [$F_{(1,41)}$ = 31.51, $p < 0.0001$], with GNX rapamycin rats showing a reduced social index versus their vehicle-treated counterparts and GIH rapamycin rats an increased social index versus their vehicle-treated counterparts (**Fig 9I and 9J**). Similar to social approach, we found that whereas rapamycin eliminated the normal preference for social novelty in GNX male offspring (**Fig 9K**), rapamycin increased the preference for social novelty in GIH male offspring (**Fig 9L**).

## Discussion

Here, we demonstrate that mimicking the exposures to IH that are characteristic of maternal SA during pregnancy results in significant neural changes in her offspring, including aberrant juvenile communicative vocalizations, increased incidence of cognitive and executive function impairment, altered social behavior, increased grooming behavior (a commonly used indicator of repetitive behavior), and structural and functional neuronal abnormalities. Interestingly, with the exception of juvenile USVs, behavior alterations were exclusively observed in GIH male, and not female, offspring. The increased preponderance of male behavioral aberrations were mirrored by pronounced prefrontal cortical spine density and morphological abnormalities in male, but not female, GIH offspring. These findings are, to our knowledge, the first direct demonstration of the effects of maternal IH during gestation on the consequent brain-based phenotypes in her juvenile, adolescent, and adult offspring, largely in a sexually dimorphic manner. Interestingly, previous nonneural studies of metabolism and cardiovascular function have also found that GIH preferentially impacts male offspring [98,99]. Our findings join a growing body of evidence indicating that insults to the maternal environment during gestation (e.g., maternal immune activation) have adverse consequences on the offspring [31,100,101]. Thus, for the first time, our data provide clear evidence that maternal SA may be an important risk factor for the development of neurodevelopmental disorders, particularly in male offspring. Further, our findings identify increased mTORC1 signaling in the mPFC of GIH male offspring and demonstrate that therapeutics aimed at dampening mTORC1 hyperactivity are able to fully rescue a broad range of behavioral deficits in these rats. These findings

further advance a growing narrative regarding the pharmacological potential of targeting mTOR signaling in certain brain-based disorders.

Several correlative human studies hint at the possibility that GIH is linked to psychiatric disorders in humans, but thus far, our understanding remains limited. Hampering large-scale efforts to investigate links between maternal SA in humans and diagnostic outcomes in her offspring is the difficulty in clearly identifying individuals who do or do not have SA. Indeed, most individuals with SA are unaware of its occurrence, and a large proportion of women without a prior history of SA transiently and unknowingly develop it during the later stages of pregnancy [102]. Further, large-scale screening for SA typically relies on questionnaires, the outcomes of which have been consistently shown as unreliable indicators of SA in the pregnant population [14–16,103]. Nevertheless, 2 small-scale studies support associations of GIH with aberrant behavioral development in the offspring. More specifically, a small sample size retrospective study found that maternal SA during pregnancy was associated with developmental vulnerability in male (but not female) children, defined as scoring below the 10th percentile in 1 or more assessment domains, including social competence, language and cognitive skills, communicative skills, and general knowledge [42]. In addition, women with confirmed SA during pregnancy that lacked any other known pregnancy complications were statistically more likely (2.5-fold increased risk) to have children who score poorly on social assessments as compared to mothers with normal gestational sleep evaluations [104]. Indeed, a recent hypothesis paper posited that maternal SA may lead to development of psychiatric disorders in the offspring, specifically autism spectrum disorders, based on observations that many of the maternal risk factors for, or consequences of, maternal SA are also risk factors for the development of autism in her offspring [41]. For example, advanced maternal age, obesity, preeclampsia, hypertension, diabetes, and maternal inflammation are all associated with maternal SA and have been linked to autism risk in the offspring [105–110]. Wardly implored investigators to begin to investigate this link; however, prior to undergoing time-consuming and expensive human studies to investigate links between maternal SA and neurodevelopmental deficits in offspring, animal models must necessarily provide the justification since ethical concerns preclude allowing a pregnancy to proceed in a mother with SA without medical intervention. To our knowledge, our report is the first to demonstrate a direct link between the IH associated with SA and offspring behavioral alterations and associated structural and functional alterations in brain regions underlying those behaviors.

Our findings indicate that male GIH offspring have deficits in spatial working memory (Y-maze) as well as impaired novelty recognition memory following a 24-hour delay, suggestive of deficits in executive function and longer-term memory maintenance, respectively. Working memory pertains to memory that is transiently relevant (on the order of minutes), under continual modification, and subject to interference. The Y-maze task requires that rats maintain and continually update a mental log of arm choice entries in order to exhibit proper alternation behavior [55]. A benefit of this task over other common assessments of working memory is the lack of need for repeated training prior to testing, the reliance on rodents' natural inclination to explore novelty, and the avoidance of exposing animals to inherent stressors (i.e., water exposure). The novel object test allows for independent assessments of the motivation to explore novelty in addition to the ability to recognize novel versus familiar stimuli. Notably, during trial 1 of the novel object test, we found that male GIH and GNX offspring showed equal amounts of time exploring objects, indicating that the deficit in novelty recognition 24 hours later is not the result of a lack of interest in object exploration, but rather, a failure to distinguish between a novel and a previously explored object.

Interacting with a conspecific is a naturally reinforcing process in rodents [111]. In the 3-chamber test, we found a deficit in social approach behavior in male GIH offspring,

indicative of a reduction in social motivation. We also found that GIH male offspring fail to show a preference for a novel versus familiar rat (preference for social novelty test). This suggests that GIH male offspring have aberrations not only in the desire to seek social interaction, but also in the more cognitive aspects of social interaction, as the social novelty preference testing requires memory for previous social encounters. One caveat in interpreting preference for social novelty is that a failure to spend more time exploring an unfamiliar versus familiar conspecific is not fully independent of changes in social motivation. Indeed, it is possible that social motivational impairments identified during the social approach task carry over into the preference for social novelty test. In this way, a lack of preference for investigating an unfamiliar versus familiar rat could manifest due not to a lack of recognition of prior social encounters per se, but rather due to a more general lack of interest in novelty engagement. However, as discussed above, our findings indicate that the inability of GIH male offspring to recognize novelty in the object recognition test was independent of the motivation to explore objects. Thus, by extrapolation between tests, it is conceivable that the preference for social novelty deficit in GIH male offspring is due to a more global reduction in the ability to recall the specifics of previous encounters, whether this be for objects or conspecifics. Nevertheless, we cannot rule out the possibility that male GIH offspring have intact social memory despite impaired social motivation.

Rodent cognitive and social behaviors depend on the proper functioning of numerous forebrain regions, including the mPFC [68–72]. Rodents do not contain a dorsolateral prefrontal cortex (DLPFC), a region critical for higher order cognition in humans and nonhuman primates. However, behavioral studies indicate that the mPFC is the closest rodent homologue of the primate DLPFC as it subsumes many of the analogous behavioral functions [68,112]. Consistent with largely male-specific deficits in cognitive and social behaviors in adult animals, we found that male, but not female, late adolescent GIH offspring exhibit robust increases in dendritic spine density in mPFC pyramidal neurons. Further, these mPFC alterations were widespread as they were not confined to a single cortical layer; they were characteristic of layer 2/3 and layer 5 pyramidal neurons. In the case of layer 2/3 neurons, we found that this increase in spine density was also present in 3-week-old male GIH offspring; however, the magnitude of increase was statistically greater at 8 weeks versus 3 weeks. Following birth and into childhood, cortical dendritic spines numbers surge, which is temporally followed by a spine pruning period that extends into adolescence and is responsible for the elimination of excess synapses [66]. The increased layer 2/3 spine density in 3-week-old male GIH offspring could be a consequence of an increase in spine formation during the early postnatal spine surge period. In male GNX offspring, we found a 16% (layer 2/3) and 20% (layer 5) reduction in spine density during the transition from 3 weeks to 8 weeks of age, consistent with spine pruning dominating during this period. On the other hand, the exacerbated increase in spine density in male GIH offspring relative to male GNX offspring during the transition from 3 weeks to 8 weeks of age is a likely consequence of a lack of layer 2/3 and layer 5 cortical spine pruning. Further, for layer 2/3, the increase in spine density in GIH male rats at 3 weeks of age suggests that there could be 2 mechanistically dissociable, yet functionally synergistic, processes contributing to the increased neuronal spine density in this cortical layer.

We assessed cortical synaptic function by assessing sEPSCs in mPFC pyramidal neurons and found an increase in sEPSC frequency, but a reduction in sEPSC amplitude, in adult male GIH offspring as compared to male GNX offspring. These sEPSC data are suggestive of an increase in the number of functional AMPAergic synapses in GIH offspring neurons (increased sEPSC frequency) and suggest that this increase does not likely come about due to an increase in the number of strong synaptic connections, but rather due to an increased preponderance of synapses with a below average strength (decreased sEPSC amplitude). While

changes in sEPSCs can arise due to presynaptic and/or postsynaptic influences, the dendritic spine alterations we detected in GIH male offspring can account for the identified sEPSC alterations. Notably, increased dendritic spine density would be expected to give rise to an increase in sEPSC frequency. Among the 3 major spine subtypes, thin spines have the smallest average spine head size. Spine head size correlates with the content of functional AMPA receptors, such that spines with large heads have a greater content of functional AMPA receptors than those with small heads [113,114]. As such, the specific increase in thin spine density in mPFC pyramidal neurons in GIH male offspring can account for the identified decrease in sEPSC amplitude.

One key question is how does GIH induce neurodevelopmental alterations in male offspring synaptic connectivity and associated behavioral dysfunction? Importantly, dams were removed from the environmental chambers prior to giving birth; thus, our observed effects are due to maternal exposure to IH and are not confounded by a perinatal hypoxic environment. Similarly, GIH did not affect gestation time, litter size, or litter survival, indicating that GIH did not have any gross effects of the health of the pregnancy. Our findings did not identify changes in HIF-1α expression or changes in oxygen sensing probe reactivity in either the placenta or in fetal brains from GIH pregnancies relative to GNX, indicating that fetal hypoxia is unlikely a major contributor to observed consequences in GIH offspring. These findings are in keeping with previous studies indicating that via the placenta, the fetus is largely protected from the effects of maternal hypoxia [44,45]. The neuronal and behavioral phenotypes observed in response to subjecting fetuses to sustained, late term prenatal hypoxia do not mirror the effects we see with GIH. Notably, sustained late term prenatal hypoxia causes a decrease in cortical spine density [115]. Moreover, subjecting pregnant rat dams to 4 hours of sustained hypoxia (10% oxygen) daily from gestation day 15 to 21 impairs cognition in 2-month-old female, but not male, offspring and actually improves cognition as compared to control offspring at 4 months of age in both males and females [116]. Further, repeated IH during gestation was shown to produce sleep deprivation and fragmentation on the first day of exposure only, with a rebound to normalcy by the second day [117]. Thus, it is not likely that maternal sleep alterations are a major contributor to the offspring phenotypes. In addition, we found normal weight growth of male and female GIH pups while in the care of their mothers during preweaning ages, suggesting that there are no gross changes in maternal care that can account for the observed phenotypes.

Rather than direct effects of hypoxia per se, we hypothesize that the maternal response to IH leads to GIH-induced alterations in neurodevelopment in offspring. Without question, IH causes profound chronic inflammation in humans and animal models that is responsible for many of the morbidities associated with SA, such as high blood pressure, heart disease, obesity, and diabetes [9–13,118]. Epidemiological and scientific studies indicate that maternal immune activation during pregnancy is a key trigger for psychiatric disorders in humans and applicable rodent models [37,119]. Maternally produced cytokines can cross the placenta and enter the fetal compartment to impact fetal brain development [120–122]. Consistent with this hypothesis, neonatal GIH offspring exhibit evidence of enhanced inflammation [123]. IH increases several cytokines, including interleukin (IL)-1β, tumor necrosis factor alpha (TNFα), IL-6, and IL-17a [11,12,124–126]. Intriguingly, in individuals with SA, the levels of IL-6 and IL-17a predict SA severity, including during pregnancy [9]; these same maternal cytokines have been most strongly linked to offspring neuropathology induced by maternal immune activation [127,128]. Future experiments will be necessary to determine the precise role for IH-induced maternal inflammation in the GIH phenotype.

Our findings do shed light on a possible biochemical mechanism in GIH male offspring that contributes to the observed aberrant phenotypes, as we find evidence for hyper mTOR

pathway signaling in the mPFC of 3-week-old GIH male offspring. Interestingly, our results implicate a specific aspect of mTOR signaling, in particular, mTORC1-mediated phosphoinhibition of ULK1. ULK1 promotes autophagy, and phosphorylation of the ULK1 S757 residue by mTOR reduces ULK1 activity and its consequent ability to facilitate autophagy [88]. Thus, the increase in phosphorylation of the ULK1 S757 residue in the mPFC of GIH male offspring is suggestive of reduced autophagy in these animals. Neuronal autophagy has many critical functions, including the pruning of superfluous dendritic spines [129]. Indeed, our previous work showed that blocking mTOR's ability to phospho-inhibit ULK1, which would expectedly increase ULK1 activity and hence increase ULK1-mediated autophagy, results in a reduction in spine density in cortical pyramidal neurons [52]. It is thus conceivable that our identified increase in ULK1 phosphoinhibition, which would expectedly decrease autophagy, contributes in part to the lack of mPFC pyramidal neuron spine pruning in GIH male offspring.

Our findings reveal that rapamycin treatment starting at 3 weeks of age and lasting until 6 weeks of age is sufficient to alleviate the cognitive and social deficits of GIH male offspring, essentially restoring performance to that of vehicle-treated GNX male offspring. Interestingly, unlike GIH male offspring, GNX male offspring did not show any improvement in cognitive or social performance with rapamycin treatment, and in fact, rapamycin adversely affected performance in these animals. This rapamycin-mediated performance deficit in the GNX males is likely due to the reduction of mTOR signaling below normal baseline levels. It would thus seem that rapamycin treatment is only beneficial in cases in which excessive levels of baseline mTOR signaling are evident. One interesting question that remains unanswered is whether there is a critical period for rapamycin treatment in alleviating the behavioral aberrations of the GIH male offspring. While we started rapamycin treatment at 3 weeks of age, it would be interesting to determine if rapamycin administration starting in young adulthood is or is not able to rescue behavioral phenotypes. Knowledge of this could help determine if the therapeutic window for rapamycin treatment in GIH male offspring remains open or closes with age.

Increased cortical dendritic spine density is a pathological hallmark of a narrow subset of neuropsychiatric disorders, namely autism spectrum disorders [77,130]. Both increased and decreased maternal separation–induced USVs have been identified in mice modeling key autism genetic susceptibility alterations [131,132], along with social withdrawal, cognitive impairments, and excessive grooming [133–135]. That male GIH offspring exhibited an increased breadth and magnitude of behavioral and dendritic spine aberrations compared to female GIH offspring is interesting given that autism is about 3-fold more prevalent in males [136]. Nevertheless, as no human epidemiological studies have attempted to examine associations of maternal gestational SA with offspring psychiatric disorders, we must exercise caution in relating our findings to any particular psychiatric condition. Further, it is possible that maternal SA produces a constellation of cognitive and social aberrations in the offspring that do not strictly fit the criteria of any one or specific psychiatric disorder. The severity and persistence of our identified phenotypes in a rodent model of maternal SA, in combination with findings from small-scale human studies and clinical correlations, highlight the value in considering a large-scale assessment in human populations in the future.

## Experimental procedures

### Experimental model and subjects details

**Experimental animals and intermittent hypoxia.** Timed pregnant Sprague–Dawley rats (E16) were obtained from Charles River (Wilmington, Massachusetts, United States of America) and housed in AAALAC-accredited facilities with 12 hour light–dark conditions. Food

and water were available to the animals ad libitum. Beginning at gestation day 10 (G10), dams were exposed to 8 h/day (9:00 AM to 5:00 PM) of IH (GIH), which consisted of alternating 2-minute hypoxic (45 seconds down to 10.5% $O_2$) and normoxic (15 seconds up to 21% $O_2$) episodes, daily for 12 days. The control group (GNX) received alternating episodes of room air (normoxia; 21% $O_2$) with the same time and gas flow parameters as GIH dams. Both groups were housed in standard rat cages with custom-made Plexiglas lids to deliver the gases. Prior to their expected delivery date (G21), lids were replaced with standard filter tops to prevent direct exposure of the pups to IH. These parameters were designed to mimic the desaturation and reoxygenation times in humans with SA. Ages of rats for individual experiments are indicated within the applicable sections below.

**Ethics statement.**   All animal experimental procedures were performed according to the NIH guidelines set forth in the Guide for the Care and Use of Laboratory Animals and with approved protocols by the University of Wisconsin–Madison Institutional Animal Care and Use Committee (protocol ID: V005173-R02). Age of animals is indicated in the applicable figure legends and is also listed in the Method details section below.

## Method details

**Behavior testing.**   With the exception of USVs that were assessed in juvenile rats (P4 and P10), GNX and GIH offspring were behaviorally tested at adolescence (4 to 7 weeks of age) and at young adulthood (12 to 17 weeks of age); different rats were used for each testing period. Within adolescent and young adult populations, age matching between GNX and GIH offspring was performed for all experiments. Testing was done on GIH and GNX offspring from a minimum of 3 mothers. While different rats were used for each testing period, in many instances, the same rats were assessed in multiple behavioral tasks, with the order from first to last as the following: Y-maze, open field, novel object recognition, social interaction, and social novelty. All animal behaviors were videotaped with an overhead digital video camera. Behavioral testing and quantification was performed by an experimenter blinded to conditions. For blinding, rats were ear tagged with arbitrary identification numbers by an investigator in a different lab than that performing the behavioral testing and analysis. This individual who was not involved in the study assigned rats to another experimenter for behavioral testing to assure that roughly equal numbers of rats per condition would be assessed for each behavioral task. The experimental group pertaining to individual identification numbers was not available to the investigator who performed the behavioral tasks and those who performed the subsequent analysis until after behavioral scoring was completed.

**Y-maze spontaneous alternation.**   Y-maze spontaneous alternation requires working memory as animals must maintain and update a mental log of arm entries. The Y-maze was purchased from Maze Engineers (Skokie, IL, USA), with each arm 11-cm wide and 50-cm long. Y-maze methods were adopted from our established methods [54,73]. For each test, criteria for arm entry included an animal having its hind paws completely within a maze arm. Rats were placed in the Y-maze, which was surrounded by distal spatial cues, and allowed free exploration. Analysis of arm entries began after the animal entered each arm once and subsequent arm choices analyzed for 10 minutes thereafter. The number of successive 3 arm alternations (clockwise or counterclockwise) was summed and then divided by the total number of arm entries minus 2. The resulting number is the spontaneous alternation percentage, with 50% being considered chance levels. The test was carried out by, and analyzed by, an experimenter blind to conditions. Any rats that did not make at least 10 arm choice during the 10-minute period were removed from analysis regardless of whether this increased or decreased chances of statistical significance (a priori criteria and done prior to unblinding).

**Novel object recognition.** Long-term memory for objects was assessed with the novel object recognition test. The recognition test was performed in a gray open field box (60-cm length × 60-cm width × 40-cm height) purchased from Maze Engineers. The stimulus objects were jars made of glass or plastic. The glass jar was clear and filled with light green plastic objects and sealed with a white cap (8.5-cm tall and 4.0-cm wide). The sides of this jar were smooth and not textured. The purple plastic jar was opaque and contained textured ripples along its side and was sealed with a blue cap (9.5-cm tall and 4.8-cm wide). The arena and objects were cleansed with 70% ethanol and thoroughly dried prior to each assessment. Preliminary investigations had ascertained that the rats showed a comparable interest in the different objects chosen for this test. Habituation: Forty-eight and 24 hours prior to testing, rats were placed in the right corner of the open field box and allowed to explore the box devoid of objects for 10 minutes. Familiarization phase: Twenty-four hours following the final habituation session, 2 identical objects were placed in opposite sides of the arena, with each object 22 cm from the respective corner of the box. Rats were placed in the center of the arena and allowed free exploration of the objects for 5 minutes, after which they were removed from the box and returned to their home cage. Testing phase: Twenty-four hours following the familiarization phase, one familiar object and one novel object were placed in the same locations as during the familiarization test. Rats were returned to the center of the arena and allowed free object exploration for 7 minutes. Rats should spend more time investigating the novel object compared to the familiar object if novel object recognition memory is intact. The novel object recognition index was calculated as the proportion of time spent investigating the novel object divided by the proportion of the total time spent investigating both objects. Any activity in which the rat directed its nose within 1 cm of a given object was considered object exploration. Other behaviors, such as using the object for rearing, attempting to climb an object (which was rare), or sitting/resting against an object, were not counted as exploratory behavior. For quantification, video recordings were slowed down to one-third of real-time speed and time spent investigating each object was quantified using a digital stopwatch with a certified resolution of 0.01 seconds and 0.0005% accuracy. This test was carried out by, and analyzed by, an experimenter blinded to conditions. Any rats that did not explore both object during the testing period or have at least 12 seconds of total object exploration were removed from analysis regardless of whether this increased or decreased chances of statistical significance (a priori criteria and done prior to unblinding).

**Social approach and social recognition.** The 3-chamber test was used to assess general sociability and recognition of social novelty. Social interaction was assessed in a gray rectangular box (80-cm length × 40.5-cm width × 40-cm height) purchased from Maze Engineers. The box was divided into 3 interconnecting chambers using a transparent partition such that the 2 end chambers were larger than the central middle chamber. A cylinder wire cage (15-cm diameter and 30-cm height), which allows for animal auditory, visual, and olfactory information transfer, was placed in the corners of each end chamber. Testing occurred in 3 sessions. Habituation session: Rats were placed in the middle chamber of an empty apparatus and allowed free exploration of the entire arena for 10 minutes. Sociability session: Twenty-four hours following habituation, sociability was assessed. During this phase, an unfamiliar rat (stimulus rat 1) was placed into a cylinder wire cage in one end chamber, while an empty cylinder wire cage was placed in the opposite end chambers. Test rats were placed in the center chamber and allowed 7 minutes of exploration of the entire arena. The end chamber with the stimulus animal was designated the social side, whereas the end chamber containing the empty wire cylinder was designated the nonsocial side. The social interaction index was calculated as the time directly investigating the cylinder in the social end chamber minus the time spent investigating the cylinder in the nonsocial end chamber. Social recognition session: Immediately following

the sociability session, a novel stranger rat (stimulus rat 2) was placed in the cylinder contained within the end chamber that had been previously empty during the sociability session. Test rats were then allowed 7 minutes of free exploration. Total exploration time of each rat-containing cylinder was assessed. All stimulus rats were treatment, age, and sex-matched to the test animals. For quantification, video recordings were slowed down to one-third of real-time speed and time spent investigating each object was quantified using a digital stopwatch with a certified resolution of 0.01 seconds and an accuracy of 0.0005%. This test was carried out by, and analyzed by, an experimenter blind to conditions. Any rats that did not enter both end chambers during the testing phases were removed from analysis regardless of whether this increased or decreased chances of statistical significance (a priori criteria and done prior to unblinding).

**Ultrasonic vocalizations.** P4 and P10 pups were separated from their mother for 15 minutes prior to testing by removing the mother from the home cage. The home cage was placed on an infrared heating pad to maintain pup body temperature. Pups were gently removed and placed in an isolation chamber made of white plastic walls. The chamber was placed inside a sound-attenuating box with an ultrasound microphone (Avisoft Bioacoustics, Glienicke/Nord-bahn, Germany) hanging from the roof of the chamber, 10 cm above the floor. Pup USVs were recorded for 5 minutes via an Avisoft UltraSoundGate 116 USB device using the Avisoft Recorder software (Avisoft Bioacoustics) with a sampling rate of 300 kHz in 16-bit format. Pups were then marked with a nontoxic surgical marker, weighed, and returned to the home cage. The chamber was cleaned with 70% ethanol and allowed to dry prior to each trial. Analysis of the first 90 seconds of elicited USV recordings was performed with Avisoft-SASLab-Pro software (Avisoft Bioacoustics) and a fast Fourier transform was conducted (512 FFT-length, 100% frame, Hamming window, 75% time window overlap). Spectrograms were produced with a 586-Hz frequency resolution and 0.4267-ms time resolution. A cutoff frequency of 25 kHz was used to exclude noise. The same rats were assessed at P4 and P10, and calls were detected using the automatic parameter measurements set up and then manually inspected and categorized by an experienced, blind observer based on previous studies [136–138].

**Open field behavior.** In the open field, locomotor behavior was semiautomatically analyzed using AnyMaze video tracking software (Wood Dale, IL, USA). After establishing the location of outer, middle, and center zones, AnyMaze automatically determines the distance traveled and time spent in each zone. During video playback in AnyMaze at 0.3 real-time speed, behaviors such as grooming and rearing are flagged by an experimenter blinded to conditions, and the program subsequently collates these parameters for analysis.

**Transcardial perfusions.** For biolistic labeling experiments, rats were transcardially perfused with PBS for 2 minutes, followed by 1.5% paraformaldehyde (PFA) in PBS for 15 minutes. Following perfusions, brains were harvested from the skull and postfixed in 1.5% PFA for 2 hours and then coronally sectioned at 150 μM on a vibratome.

**Biolistic filling of neurons in tissue slices.** Tungsten M-20 microcarriers (1.3 μm, Bio-Rad, Hercules, CA, USA) were coated with red fluorescent DiI (1,1′-dioctadecyl-3,3,3′,3′-tetra-methylindocarbocyanine perchlorate ("DiI"; DiIC18(3)). Tungsten coating was done by adding 6 mg of DiI to 100μl of dichloromethane, with vortexing. Moreover, 50 mg of tungsten was placed on a glass slide and the DiI/dichloromethane mixed and spread across the slide to create a smooth film. DiI-coated tungsten particles were scrapped from the slide into a 15-mL conical tube containing 9 mL of ultrapure water. This solution was sonicated for 10 minutes with intermittent vortexing. This solution was then inserted into PVP-coated Tefzel tubing using a 25-mL syringe, and the particles were allowed to settle for 3 minutes. The tubing was placed into a tubing preparation station (Bio-Rad) under constant rotation to ensure an even particle spread and the tubing dried via nitrogen flow. To make biolistic delivery devices, the tubing

was cut into 1-cm lengths and stored desiccated at 4°C until used. The delivery of DiI-coated tungsten particles was done using a Helios gene gun (Bio-Rad) in which helium pressure (100 psi) delivers the particles to 150-μm thick coronally sectioned brain tissue from rats that had undergone prior transcardial perfusion with 1.5% PFA. Following particle delivery, sections were stored in PBS overnight at room temperature protected from light. Sections were then incubated in 4% PFA for 1 hour at room temperature, washed 3× for 5 minutes with PBS, mounted onto glass slides, and coverslipped with hardset vectashield (Vector Laboratories, Burlingame, CA, USA).

**Dendritic spine imaging and analysis.** Second order apical dendrites in the mPFC (layers 2/3 and layers 5; infralimbic/prelimbic subregions) and second order basal dendrites in the CA3 region of the hippocampus were imaged. A total of 5 neurons (2 dendrites per neuron) were analyzed for each rat, with a minimum 45-μm length per dendrite (most between 50 and 70 μm). For spine density assessments, the $n$ used for statistical analysis and reported in the figure legend is the number of rats per condition—the data for all neurons per rat were averaged to form a single data point per animal. For spine head diameter curves, all spine head measurements across all cells, dendrites, and animals per condition were analyzed.

Dendrites were imaged using a Keyence BZ-X700E scanning microscope using a 100× objective lens at a 1.0 digital zoom. Z-stacks were obtained for each image using a 0.1-μm step size. Z-stack images were collapsed and deconvolved via a full-focus algorithm. NeuronStudio semi-automatic neuron analysis program was used for dendritic spine quantification using our established procedures [73–75]. A hollow ellipse is automatically placed on each individual spine head, such the diameter of the ellipse corresponds to the diameter of the spine head. In instances in which the ellipse over or under spans the true diameter of the spine head, manual adjustments to the ellipse diameter are made. Stubby spines were those that lacked a discernable neck. Thin and mushroom spines have a clear neck region, and spine head size was used to distinguish between these 2 subtypes. Specifically, based on the spectrum of spine head diameters within an individual brain region, spines with head diameters greater than a predetermined size (mPFC, 0.9 μm; CA3, 0.3 μm) were classified as mushroom, and those below this value as thin. These critical values were decided upon prior to the unblinding of experimental conditions. Spine imaging and analysis was done by an experimenter blind to conditions.

**Patch clamp electrophysiology recordings.** Whole-cell patch clamp experiments were performed in brain slices from layer 2/3 of the mPFC, using the slice angle and recording location. The target neurons had cell bodies approximately 300 to 400 μm from the medial pial surface, which were approximately triangular with one vertex pointing toward the pial surface. Recordings were done using a standard K-gluconate intracellular solution and standard ACSF containing 10 μM bicuculline methiodide to block GABA receptors and 25 μM AP5 to block NMDA receptors. In order to identify cell types based on their firing patterns, no TTX was used. Recorded events are technically "spontaneous" AMPA EPSCs (sEPSCs), not "miniature" AMPA EPSCs (mEPSCs). Firing patterns were classified into 5 cell types: Pyramidal (Pyr), Pyramidal Bursting (PyrBurst), Fast Spiking (FS), Single Spiking (Single), or Non-Spiking (None). A confined analysis of Pyr was performed. Basic cell properties such as input resistance (Rm), membrane capacitance (Cm), resting potential (Vr), and series resistance (Rs) were measured in each experiment. Cells were then held in current clamp to measure firing properties in response to current injections. Next, via voltage-clamp, sEPSC data (fs = 20 kHz, fc = 4 to 5 kHz, −60 mV) was collected for 5 minutes. Patch clamp experiments and analysis were done by an experimenter blind to conditions.

**Oxygen sensing hypoxyprobe and western blot.** Hypoxyprobe procedures were performed using a commercially available kit following the manufacturer's instructions (Hypoxyprobe (Burlington, MA, USA), catalog number HP1-100). Embryonic day 19 (E19) rat dams

were injected with pimonidazole (60 mg/kg, intraperitoneal). Pimonidazole is activated in hypoxic cells thereby forming adducts with protein thiol groups. Primary antibody is then used to bind and label these protein adducts in tissue homogenates. Further, 1 to 2 hours post-pimonidazole injection (one GNX/GIH cohort 1 hour, a second GNX/GIH cohort 2 hours; GIH data normalized to the GNX condition for each time point and then combined for the final analysis), dams were euthanized and fetal whole brains and placentas dissected. Tissues were flash frozen and subsequently homogenized in RIPA buffer. Total protein levels were quantified using a BCA assay (Pierce, Waltham, MA, USA) and equal protein amounts loaded onto 4–15% Tris-HCL gels (Bio-Rad) and subjected to SDS-PAGE for separation by molecular weight. Proteins were transferred onto PVDF Immobilon-P membranes (Millipore, Burlington, MA, USA) at 100 V for 1 hour in transfer buffer containing 15% methanol. Membranes were washed with Tris buffer-0.1% Tween-20, membranes blocked in 5% bovine serum albumin, and incubated with an antibody that recognizes protein thiol adducts (Mab1 antibody clone 4.311.3 from Hypoxyprobe kit, catalog number HP1-100; RRID:AB_2811309) at a dilution of 1:5,000 overnight at 4°C. Membranes with incubated with anti-mouse peroxidase secondary antibody and developed via chemiluminescence (Thermo Scientific, Waltham, MA, USA). Adduct levels were quantified using background subtracted densitometry (ImageJ) spanning the entire length of each lane. The adduct value for each sample was divided by the total amount of protein in each lane (Ponceau S).

**mPFC homogenate western blots.** mPFC tissue was microdissected from fresh coronal sections and tissue homogenized in HEPES-buffered sucrose using 20 strokes at 800 rpm with a teflon homogenizer. Equal amounts of total protein per homogenate determined using a BCA assay were resolved using 4–15% Tris-HCL gels (Bio-Rad) separated at 150 V for 2 hours. Proteins were transferred onto PVDF membranes (Millipore) in transfer buffer containing 15% methanol subjected to 100 V for 1 hour at 4°C. Membranes were subsequently dried at room temperature for at least 1 hour, rehydrated in methanol, wasted in TBS-tween 20, and then blocked with TBS-tween 20 containing 5% bovine serum albumin. Chemiluminescent substrate (Thermo Scientific) was used to develop membranes. The following primary antibodies were used: p-Akt T308 (Cell Signaling Technology (Danvers, MA, USA), Cat# 4056, RRID:AB_331163), total Akt (Cell Signaling Technology Cat# 4691, RRID:AB_915783), p-mTOR S1261 A gift from Dr. Diane Fingar of University of Michigan; antibody previously described [86], p-mTOR S2448 (Cell Signaling Technology Cat# 2971, RRID:AB_330970), total mTOR (Cell Signaling Technology Cat# 2983, RRID:AB_2105622), p-ULK1 S757 (Cell Signaling Technology Cat# 6888, RRID:AB_10829226), total ULK1 (Cell Signaling Technology Cat# 8054, RRID:AB_11178668), p-4E-BP1 T37/46 (Cell Signaling Technology Cat# 2855, RRID:AB_560835), total 4E-B1 (Cell Signaling Technology Cat# 9644, RRID:AB_2097841), p-p70S6K T389 (Cell Signaling Technology Cat# 9234, RRID:AB_2269803), total p70S6K (Cell Signaling Technology Cat# 2708, RRID:AB_390722), and actin (Thermo Fisher Scientific (Waltham, MA, USA), Cat# MA5-15739, RRID:AB_10979409).

**Rapamycin pellet administration.** Rapamycin pellets were made by Innovative Research (Sarasota, Florida, USA) with a dose of 9.45 mg per pellet with a 21-day release time. This results in a 2.5 mg/kg dose per day across the 21-day period. This pellet dose and window of administration was chosen based on a prior study [97]. The pellets were implanted in the mid-scapular region using a 7-gauge trocar under isoflurane anesthesia. Rats were given oral meloxicam (1.0 mg/kg) for pain before and 24 hours following the surgery. Pellet implantation was done at 3 weeks of age and the 21-day rapamycin release ending at 6 weeks of age. As a control, rats were implanted with a vehicle pellet. Y-maze testing was done when rats were 6 weeks old, novel object when rats were 6.6 weeks old, and social behavior when rats were 7.1 to 7.4 weeks old.

**Oxygen saturation assessment in pregnant rat dams.** Blood oxygen saturation levels in pregnant rat dams during the IH procedure was measured via pulse oximetry (STARR Life Science, Oakmonst, Pennsylvania, USA) using our previously established methods [139].

**HIF-1α analysis.** E19 dams were euthanized and fetal tissues dissected and flash-frozen tissues sonicated in Tri-Reagent (Sigma, St. Louis, Missouri, USA). Total RNA was isolated according with the manufacturer's protocols, with the inclusion of Glycoblue reagent (Invitrogen, Carlsbad, California, USA). Complementary DNA (cDNA) was synthesized from 1 ug of total RNA via MMLV reverse transcriptase and an oligo dT and random primers cocktail (Promega, Madison, Wisconsin, USA). RT-qPCR was performed using PowerSYBR green PCR mix on a StepOne system. The ddCT method was used to determine relative expression of HIF-1a to 18S ribosomal RNA. Primers were designed to span introns (Primer-BLAST, NCBI) and were purchased from Integrated DNA Technologies (Coralville, Iowa, USA):

18S forward primer: CGG-GTG-CTC-TTA-GCT-GAG-TGT-CCC
18S reverse primer: CTC-GGG-CCT-GCT-TTG-AAC-AC
HIF-1α forward primer: CTG-GAC-TTG-CCC-CTT-TCT-CTG
HIF-1α reverse primer: GGA-ACT-CAT-CCT-ATT-TTT-CTT-CTC-G.

## Quantification and statistical analysis

Data were analyzed using GraphPad Prism 8 (San Diego, CA, USA). For direct comparisons between 2 groups, 2-tailed Student $t$ tests were used. For comparison involving more than 2 groups, a 2-way ANOVA with Bonferroni or Holm–Sidak corrected post hoc was used and repeated measures included where appropriate. For USV data, following 2-way ANOVA, multiple comparison post hoc correction controlled the false discovery rate using the 2-stage linear step-up procedure of Benjamini, Krieger, and Yekutieli. Spine head diameter survival curves were deemed significantly different between groups if median head diameter survival differed by at least 5% and if comparison of the curves was statistically significant on the basis of Gehan–Breslow–Wilcoxon test. The F-test was used to gauge equal variance between groups, and Shapiro–Wilk test used to confirm normal data distribution. $n$ for individual analyses are included within the figure legends. For sEPSC comparison matrix, Kruskal–Wallis followed by Bonferroni–Holm correction for multiple comparisons was performed. ANOVA main effects and interaction statistics are included in the main text, and p and q values of $t$ tests and ANOVA post hoc tests included in the figure legends. Grubbs test was used for outlier identification and detected an outlier in the sEPSC data, as indicated in the applicable figure legends, and detected an outlier in the male GNX group for p-mTOR S1261. For all experiments, data were deemed statistically significant only for $p < 0.05$ and on graphs, $^*p < 0.05$; $^{**}p < 0.01$; $^{***}p < 0.001$.

## Supporting information

**S1 Fig. No evidence for hypoxia in GIH offspring placenta or fetus. (A)** Representative pulse oximetry trace in a pregnant rat dam in response to the cyclic oxygen fluctuations of GIH. **(B)** Graph shows the average nadir and average peak in oxyhemoglobin saturation for 5 pregnant rat dams subjected to the cyclic oxygen fluctuations of GIH. Oxyhemoglobin saturation levels reach a mean nadir of 78% at the height of hypoxia, with a mean 97% saturation level achieved between nadirs ($p = 0.0068$). $n = 5$ pregnant rat dams. **(C)** No differences in HIF-1α mRNA levels were detected between GNX and GIH offspring placentas [t (df,14) = 0.7529, $p = 0.4640$]. $n = 8$ GNX and 8 GIH placentas. **(D)** No differences in HIF-1α mRNA levels were detected between GNX and GIH offspring fetal brains [t (df,14) = 1.138, $p = 0.2744$]. $n = 8$ GNX and 8 GIH fetal brains. **(E)** Blots show levels of protein thiol adducts and total protein (Ponceau S) in GNX and GIH placentas. **(F)** Quantification revealed no differences in

protein thiol adduct levels in the placentas of GNX versus GIH offspring [t (df,14) = 0.6192, $p$ = 0.5457]. $n$ = 8 GNX and 8 GIH placentas. **(G)** Blots show levels of protein thiol adducts and total protein (Ponceau S) in GNX and GIH fetal whole brain homogenates. **(H)** Quantification revealed no differences in protein thiol adduct levels in fetal whole brain homogenates of GNX versus GIH offspring [t (df,14) = 1.114, $p$ = 0.2839]. $n$ = 8 GNX and 8 GIH fetal brains. All bar graphs are the mean + SEM. The data underlying this figure can be found in S1 Raw Data. GIH, gestational intermittent hypoxia; GNX, gestational normoxia; HIF-1α, hypoxia-inducible factor 1α.
(TIF)

**S2 Fig. USV parameters in male and female P4 and P10 GNX and GIH offspring. (A)** No differences in mean duration of individual USV call types were detected between male P4 GIH and GNX offspring. [False discovery correction post hoc (df,108): HM q = 0.9789; MS, FS, ST, CV, UP, and DN q = 1.0] $n$ = 11 GNX and 11 GIH rats. **(B)** No differences in mean duration of individual USV call types were detected between female P4 GIH and GNX offspring. [False discovery correction post hoc (df,110): HM, MS, FS q = 0.3886; ST, CV, UP, and DN q = 1.0] $n$ = 9 GNX and 13 GIH rats. **(C)** No differences in mean peak frequency of individual USV call types were detected between male P4 GIH and GNX offspring. [False discovery correction post hoc (df,108): HM, MS, FS q = 0.5148; ST q = 0.3959; CV q = 0.5148; UP q = 0.5148; DN q = 0.2196] $n$ = 11 GNX and 11 GIH rats. **(D)** No differences in mean peak frequency of individual USV call types were detected between female P4 GIH and GNX offspring. [False discovery correction post hoc (df,110): HM, MS, FS, ST, CV, UP, and DN q = 0.9430] $n$ = 9 GNX and 13 GIH rats. **(E)** No differences in mean duration of individual USV call types were detected between male P10 GIH and GNX offspring. [False discovery correction post hoc (df,65): HM, ST, MX, OT q = 0.8976] $n$ = 11 GNX and 11 GIH rats. **(F)** No differences in mean duration of individual USV call types were detected between female P10 GIH and GNX offspring. [False discovery correction post hoc (df,63): HM q = 0.8915; ST q = 0.8915; MX q = 0.6432; OT q = n/a] $n$ = 11 GNX and 12 GIH rats. **(G)** No differences in mean peak frequency of individual USV call types were detected between male P10 GIH and GNX offspring. [False discovery correction post hoc (df,65): HM, ST, MX, OT q = 0.6009] $n$ = 11 GNX and 11 GIH rats. **(H)** No differences in mean peak frequency of individual USV call types were detected between female P10 GIH and GNX offspring. [False discovery correction post hoc (df,63): HM, ST, MX q = 0.9101; OT q = n/a] $n$ = 11 GNX and 12 GIH rats. All bar graphs are the mean + SEM. The data underlying this figure can be found in S1 Raw Data. CV, chevron; DN, down; FS, frequency step harmonic; GIH, gestational intermittent hypoxia; GNX, gestational normoxia; HM, harmonic; MS, mixed syllable; P4, postnatal day 4; P10, postnatal day 10; ST, short; USV, ultrasonic vocalization.
(TIF)

**S3 Fig. Social approach activity traces. (A)** Picture of the 3-chambered arena used for social approach testing. **(B and C)** Sample traces and heat maps from 2 GNX male offspring. The traces and heat maps are from the rat's center point. **(D and E)** Sample traces and heat maps from 2 GIH male offspring. The traces and heat maps are from the rat's center point. GIH, gestational intermittent hypoxia; GNX, gestational normoxia.
(TIF)

**S4 Fig. Open field locomotor behavior in adult male GNX and GIH offspring. (A)** Schematic depicting offspring age pertaining to all data in **S4 Fig**. **(B and C)** No differences in adult GIH male offspring were detected for time [t (df,21) = 0.8508, $p$ = 0.4047] or distance traveled [t (df,21) = 0.4672 = 0.6451] in the outer zone of the open field. $n$ = 12 GNX and 11

GIH rats. **(D and E)** No differences in adult GIH male offspring were detected for time [t (df,21) = 1.261, *p* = 0.2211] or distance traveled (t (df,21) = 1.142, *p* = 0.2663] in the middle zone of the open field. *n* = 12 GNX and 11 GIH rats. **(F and G)** No differences in adult GIH male offspring were detected for time [t (df,21) = 0.7341, *p* = 0.4697] or distance traveled [t (df,21) = 0.9210, *p* = 0.3675] in the center zone of the open field. *n* = 12 GNX and 11 GIH rats. **(H and I)** No differences in adult GIH male offspring were detected for total rearing time [t (df,32) = 1.887, *p* = 0.0731] or total rearing episodes [t (df,21) = 1.345, *p* = 0.1928] in the open field. *n* = 12 GNX and 11 GIH rats. All bar graphs are the mean + SEM. The data underlying this figure can be found in S1 Raw Data. GIH, gestational intermittent hypoxia; GNX, gestational normoxia.
(TIF)

**S5 Fig. Behavioral phenotypes in adult female GNX and GIH offspring. (A)** Schematic depicting offspring age pertaining to all data in **S5 Fig**. **(B)** Analysis of mean Y-maze spontaneous alternation percentage in adult female GNX and GIH offspring revealed no differences [t (df,24) = 0.0444, *p* = 0.9649]. *n* = 12 GNX and 14 GIH rats. **(C)** Object exploration time during trial 1 of the object recognition test revealed no differences in time spent investigating either object location in GNX and GIH female offspring [Bonferroni post hoc (df,26) GNX *p* = 0.8395, GIH *p* = 1.0]. *n* = 8 GNX and 7 GIH rats. **(D)** Assessment of mean novel object recognition index revealed no differences between adult female GIH versus GNX [t (df,13) = 0.5896, *p* = 0.5656]. *n* = 8 GNX and 7 GIH rats. **(E)** Time spent investigating each cylinder in the social approach task for female adults. GIH and GNX adult female offspring spent similar time investigating the stimulus cylinder [Bonferroni post hoc (df,20) *p* = 0.6124], with no interaction of maternal GNX/GIH status by end chamber cylinder detected [F(1,10) = 0.1860, *p* = 0.6754]. **(F)** Social approach index assessment revealed no differences between adult female GIH versus GNX [t (df,10) = 0.4313, *p* = 0.6754]. *n* = 6 GNX and 6 GIH rats. **(G)** Social recognition analysis indicates that both adult female GNX and female GIH offspring spend more time investigating the novel (S2) versus familiar (S1) rat [GNX Bonferroni post hoc (df,13) *p* = 0.0010; GIH Bonferroni post hoc (df,13) *p* = 0.0183]. *n* = 8 GNX and 7 GIH rats. All bar graphs are the mean + SEM. The data underlying this figure can be found in S1 Raw Data. GIH, gestational intermittent hypoxia; GNX, gestational normoxia.
(TIF)

**S6 Fig. Open field locomotor behavior in adult female GNX and GIH offspring. (A)** Schematic depicting offspring age pertaining to all data in **S6 Fig**. **(B)** Sample locomotor traces in an open field across 10 minutes for adult female GNX and GIH offspring. Outer, middle, and inner zones of the open field are shown in orange. **(C–F)** No differences in mean speed in the open field (C), total time active (D), number of active episodes (E), or total distance traveled (F) were detected between adult female GNX and GIH offspring [(df,18), avg speed t = 0.5261, *p* = 0.6053; time active t = 1.270, *p* = 0.2201; active episodes t = 1.182, *p* = 0.2525; distance traveled t = 0.5532, *p* = 0.5869]. *n* = 10 GNX and 10 GIH rats. **(G and H)** No differences in adult GIH female offspring were detected for time [t (df,18) = 1.150, *p* = 0.2652] or distance traveled (t (df,18) = 0.9712, *p* = 0.3443) in the outer zone of the open field. *n* = 10 GNX and 10 GIH rats. **(I and J)** No differences in adult GIH female offspring were detected for time (t (df,18) = 1.179, *p* = 0.2538) or distance traveled [t(18) = 0.6521, *p* = 0.5226] in the middle zone of the open field. *n* = 10 GNX and 10 GIH rats. **(K and L)** No differences in adult GIH female offspring were detected for time [t (df,18) = 0.5125, *p* = 0.6147] or distance traveled [t (df,18) = 0.1628, *p* = 0.8725] in the center zone of the open field. *n* = 10 GNX and 10 GIH rats. **(M and N)** No differences in adult GIH female offspring were detected for total rearing time [t (df,18) = 0.9740, *p* = 0.3430] or total rearing episodes [t (df,18) = 0.1747, *p* = 0.8632] in the open field.

*n* = 10 GNX and 10 GIH rats. **(O–Q)** No differences in adult GIH female offspring were detected for total grooming time (t (df,18) = 0.2892, *p* = 0.7758), number of grooming episodes [t (df,18) = 0.1881, *p* = 0.8532], or latency to the first grooming episode [t (df,18) = 0.2531, *p* = 0.8031]. *n* = 10 GNX and 10 GIH rats. All bar graphs are the mean + SEM. The data underlying this figure can be found in S1 Raw Data. GIH, gestational intermittent hypoxia; GNX, gestational normoxia.
(TIF)

**S7 Fig. Dendritic spine head diameter assessment in layer 2/3 and layer 5 mPFC neurons from 3-week-old male GIH and GNX offspring. (A)** Schematic depicting offspring age pertaining to all data in **S7 Fig**. **(B–E)** No significant differences in layer 2/3 total or layer 2/3 spine subtype cumulative head diameter curves were identified between 3-week-old GIH and GNX offspring (total spine median survival differential 0.86%, thin differential 0.86%, stubby differential 3.2%, and mushroom differential 0.56%). *n* = 2,778 GNX and 2,636 GIH total spines; 1,712 GNX and 1,740 GIH thin spines; 603 GNX and 470 GIH stubby spines; 463 GNX and 426 GIH mushroom spines. **(F–I)** No significant differences in layer 5 total or layer 5 spine subtype cumulative head diameter curves were identified between 3-week-old GIH and GNX offspring (total spine median survival differential 3.3%, thin differential 2.3%, stubby differential 2.8%, and mushroom differential 0.36%). *n* = 1,377 GNX and 1,484 GIH total spines; 915 GNX and 935 GIH thin spines; 287 GNX and 313 GIH stubby spines; 175 GNX and 236 GIH mushroom spines. The data underlying this figure can be found in S1 Raw Data. GIH, gestational intermittent hypoxia; GNX, gestational normoxia; mPFC, medial prefrontal cortex.
(TIF)

**S8 Fig. Hippocampal CA3 field pyramidal neuron dendritic spine density and morphology in 8-week-old male GNX and GIH offspring. (A)** Schematic depicting offspring age pertaining to all data in **S8 Fig**. **(B)** Representative hippocampal CA3 field pyramidal neuron dendrite segments from 8-week-old male GNX and GIH offspring. Scale bar = 5 μm. **(C)** No significant difference in total CA3 spine density was detected between GIH and GNX offspring [t (df,4) = 2.301, *p* = 0.0829]. *n* = 3 GNX and 3 GIH rats. **(D)** CA3 spine subtype analysis revealed a significant increase in the density of thin spines [Bonferroni post hoc (df,12), *p* = 0.0130] in GIH offspring relative to GNX offspring, with no changes in the density of stubby [Bonferroni post hoc (df,12), *p* = 1.0] or mushroom spines [Bonferroni post hoc (df,12), *p* = 1.0]. *n* = 3 GNX and 3 GIH rats. **(E–H)** Spine head diameter curves for all spines (E) and for thin (F), stubby (G), and mushroom spines (H). No differences in head diameter curves between GIH and GNX offspring were detected for all spines combined or for any spine subtypes (total spine median survival differential 4.62%, thin differential 3.8%, stubby differential 3.52%, and mushroom differential 0.63%). *n* = 1,528 GNX and 1,439 GIH total spines; 1,010 GNX and 1,032 GIH thin spines; 319 GNX and 285 GIH stubby spines; 199 GNX and 122 GIH mushroom spines. All bar graphs are the mean + SEM. The data underlying this figure can be found in S1 Raw Data. GIH, gestational intermittent hypoxia; GNX, gestational normoxia.
(TIF)

**S1 Raw Images. Western blot images.**
(PDF)

**S1 Raw Data. Underlying data for figures.**
(XLSX)

## Author Contributions

**Conceptualization:** Amanda M. Vanderplow, Tracy L. Baker, Jyoti J. Watters, Michael E. Cahill.

**Data curation:** Mathew V. Jones, Michael E. Cahill.

**Formal analysis:** Amanda M. Vanderplow, Bailey A. Kermath, Mathew V. Jones, Michael E. Cahill.

**Funding acquisition:** Tracy L. Baker, Jyoti J. Watters, Michael E. Cahill.

**Investigation:** Amanda M. Vanderplow, Bailey A. Kermath, Cassandra R. Bernhardt, Kimberly T. Gums, Erin N. Seablom, Abigail B. Radcliff, Andrea C. Ewald, Mathew V. Jones, Michael E. Cahill.

**Methodology:** Amanda M. Vanderplow, Tracy L. Baker, Jyoti J. Watters, Michael E. Cahill.

**Project administration:** Michael E. Cahill.

**Resources:** Tracy L. Baker, Jyoti J. Watters, Michael E. Cahill.

**Supervision:** Michael E. Cahill.

**Validation:** Amanda M. Vanderplow, Michael E. Cahill.

**Visualization:** Michael E. Cahill.

**Writing – original draft:** Amanda M. Vanderplow, Michael E. Cahill.

**Writing – review & editing:** Amanda M. Vanderplow, Tracy L. Baker, Jyoti J. Watters, Michael E. Cahill.

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
