## [Editor Report · Decision Letter 0]

26 Apr 2021

Dear Dr Cahill, 

Thank you for submitting your manuscript entitled "A key feature of maternal gestational sleep apnea causes aberrant offspring neural phenotypes: potential implications for autism" for consideration as a Research Article by PLOS Biology.

Your manuscript has now been evaluated by the PLOS Biology editorial staff, as well as by an academic editor with relevant expertise, and I am writing to let you know that we would like to send your submission out for external peer review.

Please re-submit your manuscript within two working days, i.e. by Apr 28 2021 11:59PM.

Kind regards,

Lucas Smith, Ph.D.,

Associate Editor

PLOS Biology

---

## [Decision Letter · Decision Letter 1]

10 Jun 2021

Dear Dr Cahill,

Thank you very much for submitting your manuscript "A key feature of maternal gestational sleep apnea causes aberrant offspring neural phenotypes: potential implications for autism" for consideration as a Research Article at PLOS Biology. Your manuscript has been evaluated by the PLOS Biology editors, an Academic Editor with relevant expertise, and by several independent reviewers.

The reviews are appended below. As you will see, the reviewers think that your study is well performed, but reviewers 1 and 3 raise a number of concerns which will need to be addressed before we can consider your study for publication at PLOS Biology. Most notably, perhaps, both reviewers 1 and 3 have commented that the mechanisms for how prenatal intermittent hypoxia exerts its effects should be explored. While reviewer 1 suggests this could be done through additional discussion, having discussed this point with the Academic Editor, we think that it would be important to experimentally test (and either confirm or not) potential cellular and molecular mechanisms. For example, the Academic Editor has suggested that since you have previously shown the involvement of cytokines, especially IL-6 and IL-17a, in SA, a potentially interesting experiment might be to use neutralizing antibodies to block the effect of IL-6 and IL-17a during pregnancy and investigate if cytokines are indeed involved in the SA-related autism. We think that readouts could be USV at P4 and spine density and morphology at P21.

We appreciate that adding such mechanistic insights would require a lot of work, however we think they will significantly increase the value of this study. If it is helpful, we would be happy to discuss a revision plan with you.

As an additional note, Reviewer 1 has also commented that the 3-chambered test described here does not appear to be a standard size for use with rats (point 5). While he/she does not suggest that this data be removed or repeated, we think that in addressing this point, it would be helpful if you provide videos and/or traces of the paths taken by the rats in the chamber.

In light of the reviews, we will not be able to accept the current version of the manuscript, but we would welcome re-submission of a much-revised version that takes into account the reviewers' comments. We cannot make any decision about publication until we have seen the revised manuscript and your response to the reviewers' comments. Your revised manuscript is also likely to be sent for further evaluation by the reviewers.

We expect to receive your revised manuscript within 3 months. 

**IMPORTANT - SUBMITTING YOUR REVISION**

*Re-submission Checklist*

*Published Peer Review*

*PLOS Data Policy*

*Blot and Gel Data Policy*

Sincerely,

Lucas Smith

Associate Editor

PLOS Biology

lsmith@plos.org

REVIEWS:

Reviewer #1: This interesting manuscript addresses the intriguing hypothesis that maternal sleep apnea during pregnancy produces a risk for autism. Literature is cited for a steep rise in the incidence of sleep apnea during the third trimester of pregnancy, particularly in obese mothers, and for associations of maternal sleep apnea with psychiatric and neurodevelopmental disorders including autism in their children. 

The authors developed a rat model in which pregnant rats were exposed to intermittent hypoxia, involving cyclic bouts of 2 minutes of 10.5% oxygen, throughout 8 hours of their sleep cycle, during the second half of gestation. Several biological outcome measures were investigated: litter size and survival, gene expression of the hypoxia-inducible factor 1α (HIF-1α) in brain and placenta, dendritic spine density and morphology in medial prefrontal cortex, and AMPA-mediated spontaneous excitatory postsynaptic currents in pyramidal neurons. Behavioral consequences were tested on an appropriate sequence of rat behavioral assays relevant to the symptoms of autism spectrum disorder: ultrasonic distress vocalizations emitted by separated pups, Y-maze spontaneous alternation, novel object recognition (24 hour interval), 3-chambered sociability and social recognition. Results reveal that intermittent prenatal hypoxia led to male and female pups emitting more separation-induced vocalizations at postnatal days 4 and 10. Juveniles and adults displayed impaired cognition and social deficits, primarily in males. Further, dendritic spines densities were higher in the prenatal hypoxia groups, with differences in spine shapes. 

These studies represent an excellent set of multidisciplinary experiments across the domains of general health, pup and adult behaviors, electrophysiology, and synaptic morphology. Appropriate statistical analyses were conducted throughout. Ns of 9-13 rats per sex and per treatment group were employed. While N=9 is somewhat low for behavioral assays, results appear to be robust, and replicated at two ages. Results are well presented in clear graphs, with raw data shown in Supplementary. The Discussion section is well written in terms of summarizing the data and appropriately interpreting the results. The authors are to be congratulated on their excellent studies.

Minor changes are required to improve the manuscript:

1. Please add definitions of GIH and GNX to legends of main figure. Readers will appreciate the reminders, since the abbreviations are not self-explanatory.

2. Please add more specific descriptions of the glass and plastic novel objects, including dimensions.

3. Social approach data should be shown as time spent exploring the novel object (wire cage) and time spent exploring the novel mouse. Please add this graph and its associated statistics to the main and supplementary figures. This graphical presentation and statistical comparison are correctly shown for social recognition in Figure 2 Panel J. The same should be done for the sociability phase of the assay.

4. Note that the term "preference for social novelty" is more correct than "social recognition" for the last phase of the 3-chambered assay. Bone fide social recognition is an assay involving 4 presentations of the same partner, followed by 1 presentation of a new partner. It would be best to change to "preference for social novelty" throughout.

5. Methods for the 3-chambered test describe 15 cm diameter wire cages in an arena that measures 80 cm total length x 40 cm width. The container cage therefore is taking up approximately half the length and one third the width of its side chamber. This test usually requires more room for the subject rat to explore the entire arena, and choose to explore the novel object and the novel mouse. 

Possibly the available equipment was built for mice. For future experiments, the authors are encouraged to have a larger 3-chambered apparatus built for rats.

6. Page 27 states that the experimenter was blinded to the two treatment conditions. Please add a detailed description of exactly how the experimenter was kept uninformed of treatment condition when scoring videos.

7. References to the behavioral testing methods cite review articles rather than methods papers. Please add references to key research papers that describe the actual methods employed in the present studies, for each behavioral assay.

8. Could the authors speculate on whether the consequences of prenatal intermittent hypoxia were more likely the direct result of (a) lower oxygen during late gestation, versus (b) the presumably elevated activity of the GIH dams when awoken by apnea episodes, (c) the presumed lower amount of sleep in the awakened GIH dams, and/or (d) impairments in subsequent maternal behaviors of the GIH dams?

Reviewer #2: This seems a revised manuscript, but I was not involved in previous review nor able to download the authors' responses. As such, I can only judge this manuscript as new. The results are solid and the discussion plus conclusion is appropriate.

Reviewer #3: authors showed an interesting condition in which treatment for low oxygen of pregnant mouse will exhibit lasting effects in social behaviors in offsprings. they analyzed the social behaviors and potential cellular mechanisms involved, such as spine morphology and synaptic physiology. however, I do feel that although the data authors presented are carefully analyzed, the whole work seems too descriptive but lack of causal mechanisms. is there any molecular or cellular mechanism in any parts of brain would be account for this abnormalities? for example, is there any molecular or cellular manipulations would be able to rescue this effects caused by maternal condition? without causal connections, it is hard for this study to make important impact to the field.

---

## [Editor Report · Decision Letter 2]

29 Nov 2021

Dear Dr Cahill,

On behalf of my colleagues and the Academic Editor, Yi-Ping Hsueh, I am pleased to say that we can in principle accept your Research Article "A key feature of maternal gestational sleep apnea causes aberrant offspring neural phenotypes: potential implications for autism" for publication in PLOS Biology. Your revised manuscript has been evaluated by the PLOS Biology editorial staff and the Academic Editor, and we are satisfied by the changes made, and the substantial amount of new mechanistic insights provided. 

While we are happy to editorially accept your study, please note that before we can formally accept your manuscript and schedule it for publication, we will need you to address any remaining formatting and reporting issues, which will be detailed in an email that will follow this letter and that you will usually receive within 2-3 business days. While you wait for this email from our productions team, no action is required from you.

**Important: as you address the formatting and reporting requests, we also ask the you address the following three editorial requests:

1 - Thank you for providing, as a supplementary file, the underlying data for your study as an excel file. Will you please add a sentence to each figure legend (including supplemental figure legends) referencing this file? For example, you can add the sentence "The data underlying this figure can be found in S1 Raw_Data"

2 - Thank you for providing a supplementary file containing the raw images for your western blots. Looking at the images provided, it appears that these have been slightly cropped, which is unfortunately not compliant with our data sharing policy. Please provide the fully uncropped version of the western blot images. 

3 – Title: After discussion within the editorial team, we wonder if the title might be modified to be a bit more streamlined. If you agree, we suggest you change it to something like “Maternal sleep apnea during rat gestation causes aberrant neuronal and behavioral phenotypes in offspring”

PRESS

Sincerely, 

Lucas Smith, Ph.D. 

Senior Editor 

PLOS Biology

lsmith@plos.org